# Goal-Oriented Sequential Bayesian Experimental Design for Causal Learning

## Abstract

We present GO-CBED, a goal-oriented Bayesian framework for sequential causal experimental design. Unlike conventional approaches that select interventions aimed at inferring the full causal model, GO-CBED directly maximizes the expected information gain (EIG) on user-specified causal quantities of interest, enabling more targeted and efficient experimentation. The framework is both non-myopic, optimizing over entire intervention sequences, and goal-oriented, targeting only model aspects relevant to the causal query. To address the intractability of exact EIG computation, we introduce a variational lower bound estimator, optimized jointly through a transformer-based policy network and normalizing flow-based variational posteriors. The resulting policy enables real-time decision-making via an amortized network. We demonstrate that GO-CBED consistently outperforms existing baselines across various causal reasoning and discovery tasks—including synthetic structural causal models and semi-synthetic gene regulatory networks—particularly in settings with limited experimental budgets and complex causal mechanisms. Our results highlight the benefits of aligning experimental design objectives with specific research goals and of forward-looking sequential planning.

## 1 Introduction

A structural causal model (SCM) provides a mathematical framework for representing causal relationships via a directed acyclic graph (DAG). SCMs are foundational across domains such as genomics and precision medicine (Tejada-Lapuerta et al., 2023), economics (Varian, 2016), and the social sciences (Sobel, 2000; Imbens, 2024), where understanding the cause-effect relationships is central to scientific inquiry. Key tasks in causal modeling include: *causal discovery*, which learns the DAG structure; *causal mechanism identification*, which estimates functional dependencies; and *causal reasoning*, which answers interventional and counterfactual queries. All such tasks depend on data. While (passive) observational data can reveal correlational structures, they often fail to identify the true causal model (Verma & Pearl, 2022). In contrast, (active) *interventional* data are essential for uncovering causal relationships and estimating causal effects—but such experiments are inherently expensive and limited, making careful experimental design essential.

A Bayesian approach to optimal experimental design (BOED) (Lindley, 1956; Chaloner & Verdinelli, 1995; Rainforth et al., 2024; Huan et al., 2024) addresses this challenge by selecting interventions that maximize the expected information gain (EIG). BOED provides a principled framework for handling uncertainty in both causal structure and mechanisms. However, most existing causal BOED methods focus on *learning the full model*—for causal discovery or mechanism identification—regardless of the scientific goal.

In many real-world applications, the objective is more focused on *causal reasoning*: researchers aim to estimate the effect of a specific intervention, rather than recover the entire causal system. For instance, in drug discovery, it is often more important to understand how particular molecular targets influence disease pathways than to map the full biological network. Shown in Figure 1, optimizing for full model parameters (middle) leads to experiments that are misaligned with such targeted goals (left), resulting in inefficient use of resources (compared to right). This motivates a **goal-oriented** approach to BOED—one that tailors interventions to the specific causal queries that matter the most.

Recent work by Toth et al. (2022) begins to address goal-oriented causal design, but adopts a myopic strategy—selecting only the next experiment without planning ahead. More broadly, most causal

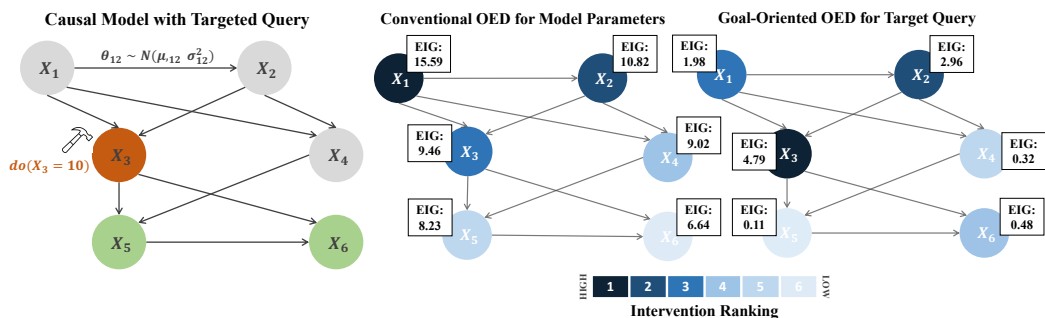

Figure 1: Illustration of goal-oriented versus conventional BOED for causal learning. *Left*: A linear Gaussian SCM with six nodes; the experimental goal is to estimate the causal effect of the intervention $do(X_3 = 10)$ on node $X_5$ and $X_6$. *Middle*: Conventional BOED selects interventions that maximize EIG over all model parameters, resulting in $X_1$ and $X_2$ being selected as the best. *Right*: Our GO-CBED approach selects interventions by directly maximizing EIG for the specific causal query, leading to a different intervention that prioritizes nodes most relevant to the query, i.e., $X_3$ and $X_2$.

BOED approaches are greedy, optimizing interventions one step at a time without accounting for how early decisions influence future learning. Overcoming this limitation requires a *non-myopic* framework, which we formulate as a Markov decision process and solve using tools from reinforcement learning (RL) (Rainforth et al., 2024, §4), (Huan et al., 2024, §5).

To address these challenges, we propose Goal-Oriented Causal Bayesian Experimental Design (**GO-CBED**), a novel framework for sequential, non-myopic causal experimental design that:

- **Directly targets user-defined causal queries**, using a variational lower-bound estimator (Poole et al., 2019; Barber & Agakov, 2004) to efficiently approximate the EIG on these specific quantities of interest (QoIs);

- **Plans non-myopically** across full experimental sequences via a learned RL policy;

- **Enables real-time intervention selection** through an amortized transformer-based policy, trained offline for fast deployment.

Our key contributions include: a **goal-oriented framework** that substantially improves experimental efficiency for specific causal queries; a **sequential, non-myopic strategy** that captures synergies between interventions; and empirical results showing that GO-CBED achieves strong performance on query-focused experimental design tasks, particularly in settings where the downstream objective involves specific causal quantities of interest.

## 2    RELATED WORK

GO-CBED builds upon and synthesizes advances from three key areas: causal BOED, goal-oriented BOED, and non-myopic sequential BOED.

Early work in causal BOED demonstrated the utility of active interventions for efficiently uncovering causal graph structures, moving beyond passive observational learning (Murphy, 2001; Tong & Koller, 2001; Cho et al., 2016; Ness et al., 2018; von Kügelgen et al., 2019; Sussex et al., 2021). Subsequent research expanded to learning full SCMs, including the selection of both intervention targets and values (Tigas et al., 2022; 2023). More recent work has highlighted the importance of tailoring experiments to specific causal QoIs (Toth et al., 2022). However, many of these approaches remain myopic—focusing on single-step gains—or are oriented toward global fidelity rather than user-specific causal objectives.

In parallel, the broader BOED literature has seen growing interest in goal-orientation design, where experiments are optimized for their utility to downstream tasks (Attia et al., 2018; Wu et al., 2021; Neiswanger et al., 2021; Huang et al., 2024; Smith et al., 2023; Zhong et al., 2024). These methods have shown substantial benefits in predictive settings, particularly with complex nonlinear models. However, they generally do not address the unique challenges of causal inference, including the interventional nature of learning and the structural constraints of SCMs.

Recognizing the limitations of greedy approaches, non-myopic sequential BOED seeks to optimize entire experimental trajectories rather than one step at a time. Approaches based on amortized policy learning and RL (Foster et al., 2021; Blau et al., 2022; Shen & Huan, 2023; Shen et al., 2025; Blau et al., 2023) have shown promise in this area. Yet in the causal setting, non-myopic strategies often focus solely on structure learning (Annadani et al., 2024; Gao et al., 2024) and do not integrate flexible, user-defined causal goals into the long-term optimization framework.

GO-CBED bridges these domains by introducing a non-myopic, goal-oriented approach to sequential experimental design in causal settings. It enables strategic planning of intervention sequences explicitly optimized to answer user-specified causal queries—such as estimating particular effects or critical mechanisms—within complex SCMs. Unlike prior methods that are goal-oriented but myopic (Toth et al., 2022) or non-myopic but focused on structure learning (Annadani et al., 2024; Gao et al., 2024), GO-CBED unifies both objectives, maximizing long-term utility for causal reasoning. A more comprehensive discussion of related work is provided in Appendix B.

## 3 PRELIMINARIES

**Structural Causal Models**  SCMs (Pearl, 2009) provide a rigorous mathematical framework for representing and reasoning about cause-effect relationships. An SCM defines a collection of random variables $\boldsymbol{X} = \{X_1, \ldots, X_d\}$, structured by a DAG $G := \{\boldsymbol{V}, E\}$. The SCM is denoted as $\boldsymbol{\mathcal{M}} := \{G, \boldsymbol{\theta}\}$, where $G$ encodes the causal structure and $\boldsymbol{\theta}$ parameterizes the causal mechanisms. Each variable $X_i$ is determined by its parents in the graph and an exogenous noise term via a structural equation:

$$X_i = f_i(\boldsymbol{X}_{\mathrm{pa}(i)}, \boldsymbol{\theta}_i; \epsilon_i), \quad \forall i \in \boldsymbol{V}. \tag{1}$$

Here, $\boldsymbol{X}_{\mathrm{pa}(i)}$ denotes the parent variables of $X_i$ in $G$, $f_i$ is a causal mechanism parameterized by $\boldsymbol{\theta}_i$, and $\epsilon_i \sim P_{\epsilon_i}$ is an independent noise variable. The SCM thus defines a joint distribution over $\boldsymbol{X}$, enabling causal reasoning and interventional analysis.

**Interventions (Experimental Designs)**  SCMs support formal reasoning about interventions—i.e., external manipulations to the system. A *perfect* (or *hard*) intervention on a subset of variables $\boldsymbol{X}_I$, denoted by $\mathrm{do}(\boldsymbol{X}_I = s_I)$ (Pearl, 2009), replaces the corresponding structural equations with fixed values $s_I$, modifying the data-generation process. This introduces an *interventional SCM*, which leads to a new distribution over the variables. Assuming causal sufficiency and independent noise (Spirtes et al., 2000), the interventional distribution follows the Markov factorization:

$$p(\boldsymbol{X}|\boldsymbol{\mathcal{M}}, \boldsymbol{\xi}) = \prod_{j \in \boldsymbol{V} \setminus I} p(X_j | \boldsymbol{X}_{\mathrm{pa}(j)}, \boldsymbol{\theta}_j, \mathrm{do}(\boldsymbol{X}_I = s_I)), \tag{2}$$

where the design variable $\boldsymbol{\xi} := \{I, s_I\}$ encodes both the intervention target $I$ and the intervention value $s_I$. Interventions form the foundation for both causal discovery (i.e., identifying $G$) and causal reasoning (i.e., estimating interventional effects).

**Goal-Oriented Sequential Bayesian Framework**  Conventional causal BOED methods typically follow a two-step procedure: first, learn the full model, and then use it to answer causal queries. Such an approach can be inefficient when only a small subset of causal QoIs matter, as it may spend significant resources learning aspects of the model irrelevant to the target queries.

To address this inefficiency, we adopt a goal-oriented perspective: rather than learning the entire model, we design experiments to directly improve our ability to answer specific causal queries. We formalize this using a query function $H$ that maps the causal model $\boldsymbol{\mathcal{M}}$ to the desired quantity $\boldsymbol{z} = H(\boldsymbol{\mathcal{M}}; \epsilon_{\boldsymbol{z}})$, where $\epsilon_{\boldsymbol{z}}$ captures any inherent stochasticity in the query. For example, setting $\boldsymbol{z} = G$ corresponds to causal discovery, while $\boldsymbol{z} = X_i^{\mathrm{do}(X_j = \psi_j)}$ corresponds to estimating the causal effect of setting $X_j = \psi_j$ on $X_i$ from a distribution of possible intervention values $\psi_j \sim p(\psi_j)$.

At experiment stage $t$ of a sequence of $T$ experiments, let the history be $\boldsymbol{h}_t := \{\boldsymbol{\xi}_{1:t}, \boldsymbol{x}_{1:t}\}$, where $\boldsymbol{\xi}_\tau$ and $\boldsymbol{x}_\tau$ denote the design and outcome of the $\tau$-th interventional experiment. The belief over the

causal model $\mathcal{M} = \{G, \boldsymbol{\theta}\}$ is updated via Bayes' rule:[1]

$$p(G|\boldsymbol{h}_t) = \frac{p(G|\boldsymbol{h}_{t-1})\,p(\boldsymbol{x}_t|G, \boldsymbol{h}_{t-1}, \boldsymbol{\xi}_t)}{p(\boldsymbol{x}_t|\boldsymbol{h}_{t-1}, \boldsymbol{\xi}_t)}, \qquad p(\boldsymbol{\theta}|G, \boldsymbol{h}_t) = \frac{p(\boldsymbol{\theta}|G, \boldsymbol{h}_{t-1})\,p(\boldsymbol{x}_t|G, \boldsymbol{\theta}, \boldsymbol{h}_{t-1}, \boldsymbol{\xi}_t)}{p(\boldsymbol{x}_t|G, \boldsymbol{h}_{t-1}, \boldsymbol{\xi}_t)},$$

where the marginal likelihood $p(\boldsymbol{x}_t|G, \boldsymbol{h}_{t-1}, \boldsymbol{\xi}_t)$ is computed by integrating over $\boldsymbol{\theta}$. However, our primary interest lies not in inferring the full model $\mathcal{M}$, but in updating beliefs about the target query $\boldsymbol{z}$. This is captured by the posterior-predictive distribution:

$$p(\boldsymbol{z}|\boldsymbol{h}_t) = \sum_G \int p(\boldsymbol{z}|G, \boldsymbol{\theta}, \boldsymbol{h}_t)\,p(G|\boldsymbol{h}_t)\,p(\boldsymbol{\theta}|G, \boldsymbol{h}_t)\,\mathrm{d}\boldsymbol{\theta}. \tag{3}$$

# 4 GOAL-ORIENTED SEQUENTIAL CAUSAL BAYESIAN EXPERIMENTAL DESIGN

**Problem Statement**  GO-CBED seeks an optimal policy $\pi : \boldsymbol{h}_{t-1} \to \boldsymbol{\xi}_t$ that maximizes the EIG on the target causal QoI $\boldsymbol{z}$ over a sequence of $T$ experiments:

$$\pi^* \in \arg\max_\pi \left\{ \mathcal{I}_T(\pi) := \mathbb{E}_{p(\mathcal{M})p(\boldsymbol{h}_T|\mathcal{M},\pi)p(\boldsymbol{z}|\mathcal{M})}\left[ \log \frac{p(\boldsymbol{z}|\boldsymbol{h}_T)}{p(\boldsymbol{z})} \right] \right\}, \tag{4}$$

subject to the constraint that designs following the policy: $\boldsymbol{\xi}_t = \pi(\boldsymbol{h}_{t-1})$ for all $t$. This formulation is *goal-oriented*, as it directly targets EIG on specific causal QoIs rather than the full model, and *non-myopic*, as it optimizes the entire sequence of interventions rather than selecting each greedily. An equivalent formulation based on incremental (stage-wise) EIG after each experiment is also possible; see Appendix A.1 for details.

The EIG on $\boldsymbol{z}$ defined in equation 4, $\mathcal{I}_T$, is also the mutual information between $\boldsymbol{z}$ and $\boldsymbol{h}_T$. When $\boldsymbol{z}$ is a bijective function of $\mathcal{M}$, maximizing EIG on $\boldsymbol{z}$ is equivalent to maximizing it on $\mathcal{M}$ (Bernardo, 1979). However, when $\boldsymbol{z}$ is not invertible with respect to $\mathcal{M}$, directly maximizing EIG on $\boldsymbol{z}$ is more efficient, avoiding effort on irrelevant parts of $\mathcal{M}$ and reducing both computational and experimental costs—especially beneficial when dealing with large causal graphs or tight intervention budgets.

## 4.1 VARIATIONAL LOWER BOUND

Evaluating and optimizing the EIG in equation 4 requires estimating the posterior density $p(\boldsymbol{z}|\boldsymbol{h}_T)$, which is generally intractable for complex causal models. To address this, we adopt a *variational approach* that approximates the posterior using $q_{\boldsymbol{\lambda}}(\boldsymbol{z}|\pi, f_{\boldsymbol{\phi}}(\boldsymbol{h}_T))$, where $\boldsymbol{\lambda}$ is the variational parameter and $f_{\boldsymbol{\phi}}$ is a learned embedding of the historical interventional data:

$$\mathcal{I}_{T;\,L}(\pi; \boldsymbol{\lambda}, \boldsymbol{\phi}) := \mathbb{E}_{p(\mathcal{M})p(\boldsymbol{h}_T|\mathcal{M},\pi)p(\boldsymbol{z}|\mathcal{M})}\left[ \log \frac{q_{\boldsymbol{\lambda}}(\boldsymbol{z}|f_{\boldsymbol{\phi}}(\boldsymbol{h}_T))}{p(\boldsymbol{z})} \right], \tag{5}$$

subject to $\boldsymbol{\xi}_t = \pi(\boldsymbol{h}_{t-1})$ for all $t$.

**Theorem 4.1** (Variational Lower Bound). *For any policy $\pi$, variational parameter $\boldsymbol{\lambda}$, and embedding parameter $\boldsymbol{\phi}$, the EIG satisfies $\mathcal{I}_T(\pi) \geq \mathcal{I}_{T;\,L}(\pi; \boldsymbol{\lambda}, \boldsymbol{\phi})$. The bound is tight if and only if $p(\boldsymbol{z}|\boldsymbol{h}_T) = q_{\boldsymbol{\lambda}}(\boldsymbol{z}|f_{\boldsymbol{\phi}}(\boldsymbol{h}_T))$ for all $\boldsymbol{z}$ and $\boldsymbol{h}_T$.*

A proof is provided in Appendix A.2. Since $p(\boldsymbol{z})$ is independent of $\pi$, $\boldsymbol{\lambda}$, and $\boldsymbol{\phi}$, it can be omitted from the optimization statement without affecting the argmax. Thus, maximizing the EIG lower bound reduces to maximizing the *prior-omitted* EIG bound:

$$\pi^*, \boldsymbol{\lambda}^*, \boldsymbol{\phi}^* \in \arg\max_{\pi, \boldsymbol{\lambda}, \boldsymbol{\phi}} \left\{ \mathcal{R}_{T;L}(\pi; \boldsymbol{\lambda}, \boldsymbol{\phi}) := \mathbb{E}_{p(\mathcal{M})p(\boldsymbol{h}_T|\mathcal{M},\pi)p(\boldsymbol{z}|\mathcal{M})}\left[ \log q_{\boldsymbol{\lambda}}(\boldsymbol{z}|f_{\boldsymbol{\phi}}(\boldsymbol{h}_T)) \right] \right\}, \tag{6}$$

where $\mathcal{R}_{T;L}(\pi; \boldsymbol{\lambda}, \boldsymbol{\phi}) \leq \mathcal{R}_T(\pi) := \mathbb{E}_{p(\mathcal{M})p(\boldsymbol{h}_T|\mathcal{M},\pi)p(\boldsymbol{z}|\mathcal{M})}[\log p(\boldsymbol{z}|\boldsymbol{h}_T)]$ is a lower bound to the prior-omitted EIG $\mathcal{R}_T(\pi)$. See Appendix A.3 for additional information on $\mathcal{R}_{T;L}(\pi; \boldsymbol{\lambda}, \boldsymbol{\phi})$.

---

[1] When observational data $\mathcal{D}$ is available prior to designing interventions, all distributions are implicitly conditioned on $\mathcal{D}$. See Appendix C.5 for further details.

## 4.2 Variational Posteriors and Policy Network

Having established the theoretical foundation of GO-CBED, we now describe its implementation. Our approach comprises two key components: (1) variational posteriors for establishing the EIG lower bound, and (2) a policy network that guides the intervention selection process.

**Variational Posteriors**  While GO-CBED supports arbitrary causal queries, we focus on two fundamental tasks that form the basis of our experimental evaluation: causal reasoning (i.e., estimating interventional effects) and causal discovery (i.e., learning graph structure).

For causal reasoning tasks, where the query takes the form $z = X_i^{\text{do}(X_j = \psi_j)}$, the posterior distribution $p(z|h_T)$ is often complex and multimodal due to the structural uncertainty—different causal graphs can imply different causal effects for the same intervention. To capture this complexity, we parameterize the variational posterior $q_\lambda(z|f_\phi(h_T))$ using normalizing flows (NFs), which transform a Gaussian base distribution into a flexible target distribution via a series of invertible mappings, while enabling efficient density estimation. Specifically, we use the Real NVP architecture (Dinh et al., 2016) and follow the implementation strategy of Dong et al. (2025). Details are provided in Appendix A.4.

For causal discovery tasks, where the query is the graph itself, $z = G$, we model the posterior over graph structures using an independent Bernoulli distribution for each potential edge (Lorch et al., 2022):

$$q_\lambda(G|f_\phi(h_T)) = \prod_{i,j} q_\lambda(G_{i,j}|f_\phi(h_T)), \tag{7}$$

where each $q_\lambda(G_{i,j}|\cdot) \sim \text{Bernoulli}(\lambda_{i,j})$. This parameterization allows efficient modeling of the posterior over DAG structures, while maintaining scalability and differentiability.

**Policy Network**  We represent the policy $\pi$ using a neural network with parameters $\gamma$. The policy network selects the next intervention by mapping the history $h_{t-1}$ to a design $\xi_t$ at stage $t$. The architecture is designed to satisfy two symmetry properties that have been shown to improve performance Annadani et al. (2024): permutation invariance across history samples and permutation equivariance across variables.

The network is composed of $L$ transformer layers that alternate between attention over variable and observation dimensions. This alternating structure enables rich and efficient information flow across the entire history of interventions and outcomes. The final embedding is passed through two output heads: one that produces the intervention targets $I_t$, using the Gumbel-softmax trick to enable differentiability for discrete variables, and the other predicts the corresponding intervention values $s_{I_t}$. The architecture is illustrated in Appendix C.1.

---

**Algorithm 1** The GO-CBED algorithm.

---

1: **Input**: $H$, $p(\psi)$; prior $p(G)$, $p(\theta|G)$; likelihood $p(x_t|G, \theta, \xi_t)$; number of experiments $T$;
2: Initialize policy network parameters $\gamma$, variational parameters $\lambda$, embedding parameters $\phi$;
3: **for** $l = 1, \ldots, n_{\text{step}}$ **do**
4:     Simulate $n_{\text{env}}$ samples of $G$, $\theta$, $\psi$ and $z$;
5:     **for** $t = 0, \ldots, T$, **do**
6:         Compute $\xi_t = \pi(h_{t-1})$, then sample $x_t \sim p(x_t|G, \theta, \xi_t)$;
7:     **end for**
8:     Update $\gamma$, $\lambda$, and $\phi$ following gradient ascent, where gradient obtained from auto-grad on $\mathcal{R}_{T;L}$;
9: **end for**
10: **Output**: Optimal $\pi^*$ parameterized by $\gamma^*$, and $\lambda^*$, and $\phi^*$;

---

**Training Procedure**  Algorithm 1 outlines our training procedure, which jointly optimizes the policy parameters and variational parameters by maximizing the variational lower bound. In each training iteration, we sample causal models $\mathcal{M} = (G, \theta)$ from the prior, derive the target QoIs $z$, simulate intervention trajectories, and update all network parameters via gradient ascent. At deployment time, only forward passes through the policy network are required, eliminating the need for online Bayesian inference. This enables real-time decision-making with constant computational complexity, independent of experiment sequence length.

## 5 NUMERICAL RESULTS

Our numerical experiments demonstrate how GO-CBED advances causal experimental design through goal-oriented optimization. We begin in Section 5.1 to recap the motivating example from Figure 1 that illustrates the fundamental advantage of targeting specific causal queries over full model learning. We then focus on two key causal tasks: Section 5.2 examines causal reasoning, where we evaluate performance in estimating targeted causal effects across both synthetic causal models and semi-synthetic gene regulatory networks derived from the Dialogue for Reverse Engineering Assessments and Methods (DREAM) benchmarks (Greenfield et al., 2010); Section 5.3 then turns to causal discovery, comparing GO-CBED against existing causal BOED baselines on similar synthetic and semi-synthetic settings.

### 5.1 MOTIVATING EXAMPLE WITH FIXED GRAPH STRUCTURE

We first evaluate the benefits of goal-oriented policies on the motivating example with a fixed graph structure in Figure 1. This setup assumes a linear-Gaussian relationship between variables, allowing for analytical posterior computation and accurate EIG estimation. We compare four policies: **GO-CBED-$z$**, which is optimized for the specific causal query; **GO-CBED-$\theta$**, which targets model parameters; **NMC**, a baseline that uses the nested Monte Carlo (NMC) estimator for the prior-omitted EIG on QoIs (Toth et al., 2022); and **Random**, which selects both intervention targets and values uniformly at random. We evaluate their performance on the prior-omitted EIG (or lower bound) for $z$. Full experiment details can be found in Appendix C.2.

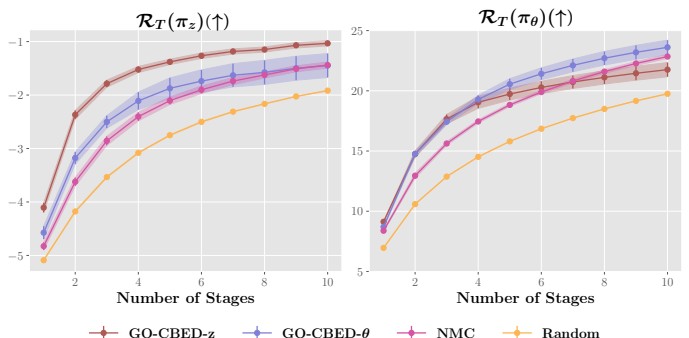

Figure 2: Performance comparison of policies trained for $T = 10$, evaluated across different stage lengths. *Left*: Performance on causal query $z = \{X_5, X_6 \mid \mathrm{do}(X_3 = 10)\}$. *Right*: Performance on model parameters $z = \{\theta \setminus \theta_{\mathrm{pa}(3)}\}$. While GO-CBED-$\theta$ achieves higher EIG on the task centering parameters (right), it performs significantly worse than GO-CBED-$z$ on the causal reasoning task (left). Shaded regions represent $\pm 1$ standard error across 4 random seeds.

Figure 2 reveals a key insight: although GO-CBED-$\theta$ achieves higher EIG on model parameters, its performance on the actual causal query is substantially worse than that of GO-CBED-$z$. This supports our central argument—when the goal is to answer specific causal queries, policies that directly target those queries are significantly more efficient than those optimized for general model learning. Moreover, GO-CBED's variational formulation consistently outperforms the sampling-based NMC. This advantage is especially pronounced when the inner-loop sample size in NMC is small, where the estimator suffers from high bias (see Appendix D.1).

### 5.2 CAUSAL REASONING TASKS

We evaluate GO-CBED's ability to design interventions that maximize EIG with respect to specific causal queries, now no longer fixing the graph structure. We compare three policies: **GO-CBED-$z$** optimized directly for causal queries, **GO-CBED-$G$** trained for causal discovery, and **Random** selection. Additional experiment details are provided in Appendix C.3.

**Metrics** We evaluate each method on both the prior-omitted EIG lower bound $\mathcal{R}_{T;L}$ and the downstream performance, measured by the **Wasserstein Distance (WD)** between the ground-truth predictive distribution $p(z \mid G^*, \theta^*)$ and the policy-specific learned posterior $q_\lambda(z \mid f_\phi(\cdot))$, obtained with a trajectory simulated from each policy.

**Synthetic SCMs** Figure 3 compares performance using Erdös–Rényi (ER) and Scale-free (SF) graph priors with nonlinear mechanisms and $d = 10$, detailed in Appendix C.3. Across all cases,

GO-CBED-$z$ outperforms the other methods significantly on both policy quality and downstream prediction. Despite the strong performance of GO-CBED-$G$ on causal discovery tasks (see Appendix C.3), these results highlight a key insight: for complex causal mechanisms, directly targeting causal queries is particularly advantageous compared to targeting the full causal graph. Moreover, for nonlinear mechanisms parameterized by neural networks, the dimensionality of the weights $\theta$ is large and graph-dependent, which makes it difficult to generate sufficient samples to effectively tighten the EIG lower bound over full graphs and parameters. These challenges underscore the advantages that goal-oriented design is intended to provide.

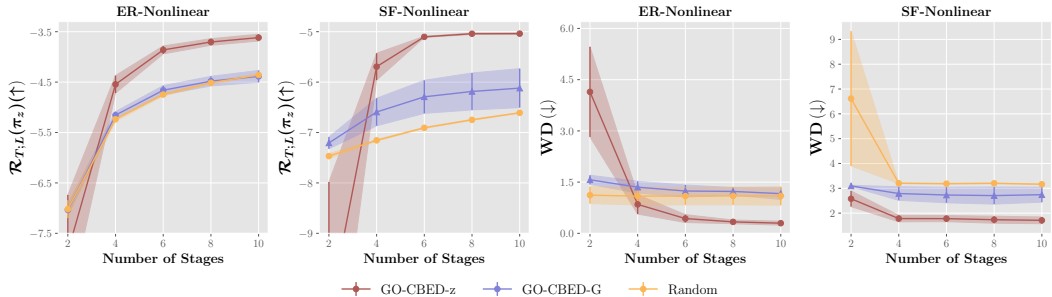

Figure 3: **Causal Reasoning on Synthetic SCMs.** Performance comparison of policies trained for $T = 10$ on causal queries, using ER and SF graph priors with nonlinear mechanisms (3 interventions per stage, $d = 10$). While GO-CBED-$G$—which targets structure learning—is a natural baseline, it underperforms on causal queries, particularly in nonlinear settings. In contrast, GO-CBED-$z$, which directly targets causal QoI, consistently achieves better performance. Shaded regions represent $\pm 1$ standard error over 4 random seeds.

**Semi-Synthetic Gene Regulatory Networks**  To assess real-world applicability, we evaluate GO-CBED on semi-synthetic gene regulatory networks derived from the DREAM (Greenfield et al., 2010) benchmarks, detailed in Appendix C.3. Figure 4 presents results on *E. coli* networks with nonlinear causal mechanisms ($d = 10$, $T = 10$). GO-CBED-$z$, which directly targets causal query, significantly outperforms both GO-CBED-$G$ and Random baselines, especially after the initial stages of intervention. This performance gap on biologically-inspired networks has important practical implications. In real biological research, experimental resources are often limited, and researchers

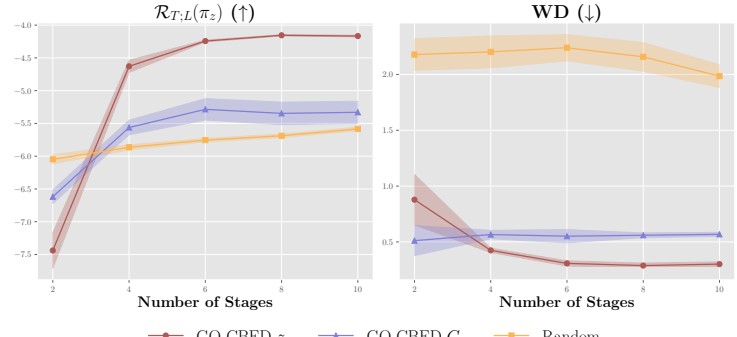

Figure 4: **Causal Reasoning on Semi-Synthetic GRNs.** Performance comparison of policies trained for $T = 10$ on *E. coli* gene regulatory networks with nonlinear causal mechanisms ($d = 10$). GO-CBED-$z$ performs comparably to baselines in early stages but exhibits rapid improvement after stage 3, ultimately achieving substantially higher prior-omitted EIG lower bound than baselines. These results highlight the value of goal-oriented experimental design in realistic biological settings with complex nonlinear causal mechanisms. Shaded regions represent $\pm 1$ standard error across 4 random seeds.

typically seek to answer specific causal questions rather than infer the entire network structure. GO-CBED's ability to efficiently target such queries highlights its promise for applications such as gene regulatory network analysis and drug target identification, where maximizing information about specific causal effects is critical. We further validate GO-CBED's effectiveness through additional experiments on Yeast networks and with diverse goal specifications in Appendix D.2. Additional evaluations of GO-CBED's robustness to distributional shifts in observation noise are presented in Appendix D.5.

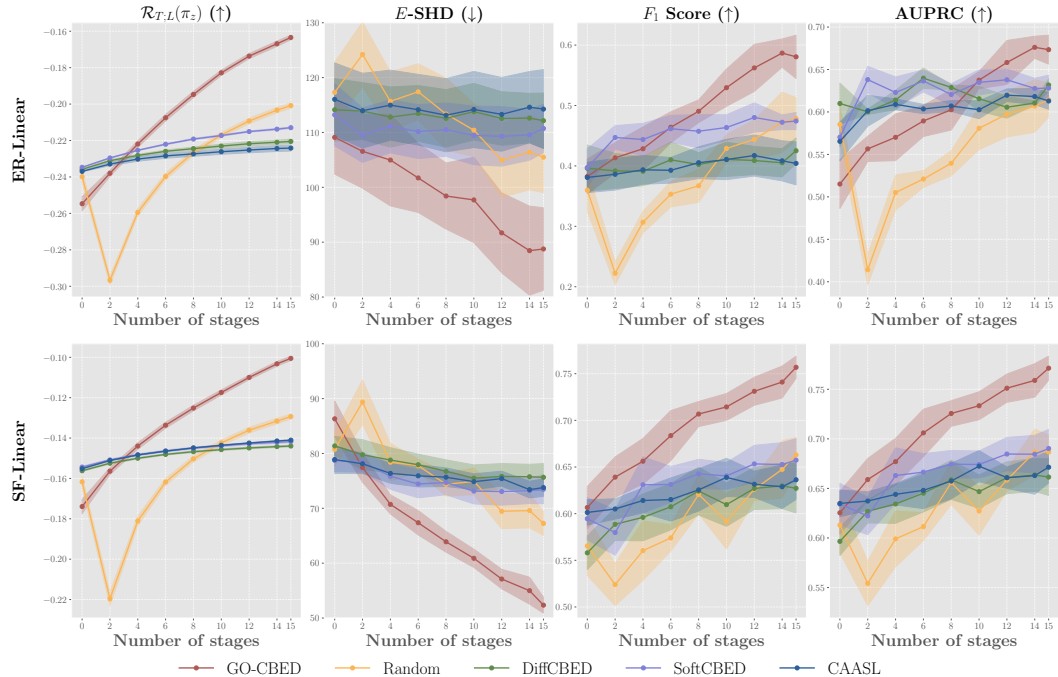

Figure 5: **Causal Discovery on Synthetic SCMs.** Performance comparison on synthetic SCMs, using ER and SF graph priors with linear mechanisms ($d = 30, T = 10$). Metrics include prior-omitted EIG lower bound $\mathcal{R}_{T;L}$, expected structural Hamming distance $\mathbb{E}$-SHD, $F_1$-score, and AUPRC. GO-CBED performs better in terms of uncertainty reduction ($\mathcal{R}_{T;L}$), structural recovery ($\mathbb{E}$-SHD), and structural accuracy ($F_1$-score and AUPRC) compared to all baselines. Shaded regions indicate $\pm 1$ standard error across 10 random seeds.

## 5.3 CAUSAL DISCOVERY TASKS

While GO-CBED is primarily designed for general goal-oriented experimental design, we also apply it to specific causal discovery tasks, where the target QoI is the causal graph $z = G$. This enables comparison with existing causal BOED methods specifically designed for structure learning. We consider synthetic settings using ER and SF graph priors, and semi-synthetic settings based on real gene regulatory networks from DREAM (Greenfield et al., 2010). In all cases, we simulate linear and nonlinear causal mechanisms with additive noise. See Appendix C.3 for more details.

**Baselines** We benchmark GO-CBED against four methods: **CAASL** (Annadani et al., 2024), which uses an offline RL method with a fixed pre-trained posterior network; **Random**, which uniformly selects interventions at random; **Soft-CBED** (Tigas et al., 2022), which employs Bayesian optimization for single-step EIG; and **DiffCBED** (Tigas et al., 2023), which learns a non-adaptive policy through gradient-based optimization.

**Metrics** We evaluate using four metrics: prior-omitted EIG lower bound $\mathcal{R}_{T;L}$; expected structural Hamming distance $\mathbb{E}$-SHD (de Jongh & Druzdzel, 2009) between posterior graph samples and the ground truth; and, for edge prediction, $F_1$-score and area under the precision–recall curve (AUPRC). To ensure a fair comparison across policies, we train a dedicated posterior network for each policy. This isolates the contribution of the policy itself and avoids confounding effects from differing posterior approximation methods. For example, while CAASL relies on a fixed pre-trained posterior network from (Lorch et al., 2022), other baselines use DAG-bootstrap (Friedman et al., 2013; Hauser & Bühlmann, 2012) for linear SCMs and DiBS (Lorch et al., 2021) for nonlinear SCMs. In our evaluation, we adopt posterior networks for inference across all baselines, as they produce higher-quality posteriors than those used in the original works. For completeness, results using each method's original inference setup are included in Appendix D.4. All results are averaged over 10 random seeds.

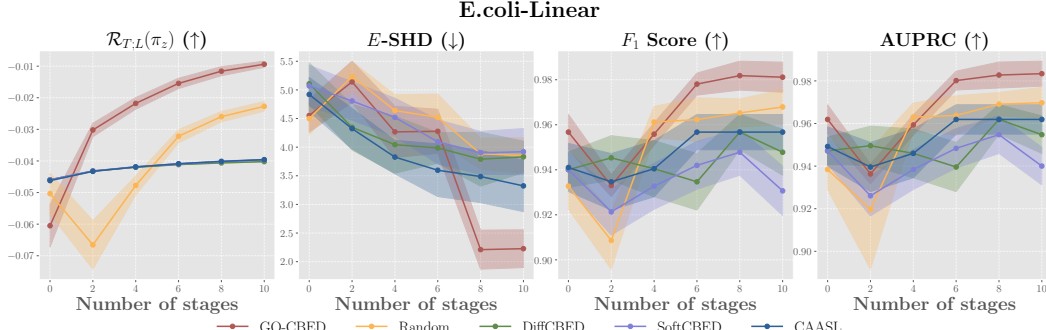

Figure 6: **Causal Discovery on Semi-Synthetic GRNs.** GO-CBED outperforms all baselines on semi-synthetic *E. coli* gene regulatory networks ($d = 20$, $T = 10$) with linear mechanisms. Our method achieves near-zero $\mathcal{R}_{T;L}(\pi_{\boldsymbol{z}}; \boldsymbol{\lambda}, \boldsymbol{\phi})$, significant lower $\mathbb{E}$-SHD, superior $F_1$ score and AUPRC for edge prediction. This demonstrates GO-CBED's ability to efficiently identify the true causal structure in biologically-inspired networks, with the variational posterior tightly concentrated around the ground truth after 10 stages.

**Synthetic SCMs**   Figure 5 presents GO-CBED's performance using ER and SF graph priors with linear mechanisms ($d = 30$, $T = 15$). Across all metrics and graph types, GO-CBED consistently outperforms baseline methods, with the gap widening as the number of stages increases. On SF graphs, GO-CBED reaches $F_1 \approx 0.75$ and attains the highest AUPRC, indicating a stronger precision-recall trade-off in sparse settings. Although GO-CBED initially performs comparably to some baselines, it steadily surpasses them as more interventions are collected. This highlights its strength in optimizing long-term information gain rather than short-term or greedy improvements. While the advantage is most prominent in linear settings, GO-CBED still achieves strong performance in the more challenging nonlinear cases, with results provided in Appendix D.3.

**Semi-Synthetic Gene Regulatory Networks**   Figure 6 evaluates GO-CBED on 20-node networks derived from the DREAM *E. coli* gene regulatory benchmark with linear mechanisms. Our method consistently outperforms all baselines across all evaluation metrics. By the final intervention stage, GO-CBED achieves high $\mathcal{R}_{T;L}$ values, low $\mathbb{E}$-SHD scores, high $F_1$-scores and AUPRC, indicating accurate recovery of the true causal structure with minimal posterior uncertainty. This strong performance on semi-synthetic networks with realistic biological topologies demonstrates GO-CBED's ability to handle the complex dependencies and noise typically encountered in gene regulatory systems. Additional experiments on Yeast networks and with nonlinear mechanisms presented in Appendix D.3 further support GO-CBED's effectiveness in biologically relevant settings.

## 6   DISCUSSION

We presented GO-CBED, a goal-oriented Bayesian framework for sequential causal experimental design. Unlike conventional approaches that aim to learn the full causal model, GO-CBED directly maximizes the EIG on specific causal QoIs, enabling more targeted and efficient experimentation. The framework is both non-myopic, optimizing over entire sequences of interventions, and goal-oriented, focusing on model aspects relevant to the causal query. To overcome the intractability of exact EIG computation, we introduced a variational lower bound, optimized jointly over policy and variational parameters. Our implementation leveraged NFs for flexible posterior approximations and a transformer-based policy network that captures symmetry and structure in the intervention history. Numerical experiments demonstrated that GO-CBED outperforms baseline methods in multiple causal tasks, with gains increasing as causal mechanisms become more complex. Crucially, the joint training of intervention policies and variational posteriors enabled adaptive, goal-oriented exploration of the causal model.

**Limitations and Future Work**   While GO-CBED demonstrates strong empirical performance, several limitations remain. Its scalability is constrained by the complexity of both the underlying causal models and the neural network architectures. Additionally, its effectiveness depends on the availability of prior knowledge of causal structures and mechanisms. Future work includes incorporating foundation models as high-fidelity world simulators for offline policy training. Recent advances in biological foundation models (Theodoris et al., 2023; Cui et al., 2024) offer a promising avenue

simulating complex, realistic causal mechanisms, which could significantly enhance policy learning without relying on costly real-world experimentation. Other valuable extensions include generalizing GO-CBED to support multi-target intervention settings and non-differentiable likelihoods, as well as improving policy robustness to changing experimental horizons and dynamic model updates during experimentation.

## 7    Ethics Statement

Our work presents a methodological contribution to causal experimental design with applications in scientific domains such as biological research. While our proposed framework is designed to improve experimental efficiency for beneficial tasks, we acknowledge that causal inference methods can have dual-use implications and could potentially be misused to identify harmful causal pathways. We are committed to promoting reproducible research by making our implementation available and encourage users to implement appropriate privacy protections when applying our methods to sensitive domains.

## 8    Reproducibility Statement

To ensure reproducibility, we provide experimental details in the appendix, including all hyperparameter configurations and data generation procedures (Appendix C). Theoretical derivations and proofs are available in Appendix A. The source code for our proposed framework and experimental setups is included in the supplementary materials and will be made publicly available upon acceptance.

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

APPENDIX

## A  Theoretical and Numerical Formulations

### A.1  Incremental EIG Formulation

The incremental EIG on the target query $\boldsymbol{z}$ resulting from an experiment at stage $t$ with design $\boldsymbol{\xi}_t$, given the intervention history $\boldsymbol{h}_{t-1}$, is defined as:

$$\mathcal{I}_t(\boldsymbol{\xi}_t, \boldsymbol{h}_{t-1}) := \mathbb{E}_{p(\boldsymbol{\mathcal{M}}|\boldsymbol{h}_{t-1})p(\boldsymbol{x}_t|\boldsymbol{\mathcal{M}},\boldsymbol{\xi}_t)p(\boldsymbol{z}|\boldsymbol{\mathcal{M}})} \left[ \log \frac{p(\boldsymbol{z}|\boldsymbol{h}_t)}{p(\boldsymbol{z}|\boldsymbol{h}_{t-1})} \right]. \tag{A1}$$

**Proposition A.1.** *The total EIG of a policy $\pi$ on the target query $\boldsymbol{z}$ over a sequence of $T$ experiments can be written as:*

$$\mathcal{I}_T(\pi) = \mathbb{E}_{p(\boldsymbol{h}_T|\pi)} \left[ \sum_{t=1}^T I_t(\boldsymbol{\xi}_t, \boldsymbol{h}_{t-1}) \right]. \tag{A2}$$

*Proof.* Beginning from the right-hand side, we have:

$$\begin{aligned}
&\mathbb{E}_{p(\boldsymbol{h}_T|\pi)} \left[ \sum_{t=1}^T \mathcal{I}_t(\boldsymbol{\xi}_t, \boldsymbol{h}_{t-1}) \right] \\
&= \mathbb{E}_{p(\boldsymbol{h}_T|\pi)} \left[ \sum_{t=1}^T \mathbb{E}_{p(\boldsymbol{\mathcal{M}}|\boldsymbol{h}_{t-1})p(\boldsymbol{x}_t|\boldsymbol{\mathcal{M}},\boldsymbol{\xi}_t)p(\boldsymbol{z}|\boldsymbol{\mathcal{M}})} \left[ \log \frac{p(\boldsymbol{z}|\boldsymbol{h}_t)}{p(\boldsymbol{z}|\boldsymbol{h}_{t-1})} \right] \right] \\
&= \sum_{t=1}^T \left[ \mathbb{E}_{p(\boldsymbol{h}_{t-1}|\pi)p(\boldsymbol{\mathcal{M}}|\boldsymbol{h}_{t-1})p(\boldsymbol{x}_t|\boldsymbol{\mathcal{M}},\boldsymbol{\xi}_t)p(\boldsymbol{z}|\boldsymbol{\mathcal{M}})} \left[ \log \frac{p(\boldsymbol{z}|\boldsymbol{h}_t)}{p(\boldsymbol{z}|\boldsymbol{h}_{t-1})} \right] \right] \\
&= \sum_{t=1}^T \left[ \mathbb{E}_{p(\boldsymbol{\mathcal{M}},\boldsymbol{x}_t,\boldsymbol{h}_{t-1},\boldsymbol{z}|\pi)} \left[ \log \frac{p(\boldsymbol{z}|\boldsymbol{h}_t)}{p(\boldsymbol{z}|\boldsymbol{h}_{t-1})} \right] \right] \\
&= \sum_{t=1}^T \left[ \mathbb{E}_{p(\boldsymbol{\mathcal{M}},\boldsymbol{h}_t|\pi)p(\boldsymbol{z}|\boldsymbol{\mathcal{M}})} \log p(\boldsymbol{z}|\boldsymbol{h}_t) - \mathbb{E}_{p(\boldsymbol{\mathcal{M}},\boldsymbol{h}_{t-1}|\pi)p(\boldsymbol{z}|\boldsymbol{\mathcal{M}})} \log p(\boldsymbol{z}|\boldsymbol{h}_{t-1}) \right] \\
&= \mathbb{E}_{p(\boldsymbol{\mathcal{M}},\boldsymbol{h}_T|\pi)p(\boldsymbol{z}|\boldsymbol{\mathcal{M}})} \left[ \log \frac{p(\boldsymbol{z}|\boldsymbol{h}_T)}{p(\boldsymbol{z})} \right] \\
&= \mathcal{I}_T(\pi), \tag{A3}
\end{aligned}$$

where in the third equality, the joint expectation is formed using $p(\boldsymbol{z}|\boldsymbol{\mathcal{M}}) = p(\boldsymbol{z}|\boldsymbol{\mathcal{M}}, \boldsymbol{h}_{t-1})$, and the fifth equality follows from the cancellation of all terms in the summation except the first and last.

$\square$

## A.2 PROOF FOR THEOREM 4.1

*Proof for Theorem 4.1.* Following equation 4 and equation 5, the difference

$$\mathcal{I}_T(\pi) - \mathcal{I}_{T;L}(\pi; \boldsymbol{\lambda}, \boldsymbol{\phi})$$

$$= \mathbb{E}_{p(\boldsymbol{\mathcal{M}})p(\boldsymbol{h}_T|\boldsymbol{\mathcal{M}},\pi)p(\boldsymbol{z}|\boldsymbol{\mathcal{M}})}\left[\log\frac{p(\boldsymbol{z}|\boldsymbol{h}_T)}{p(\boldsymbol{z})}\right] - \mathbb{E}_{p(\boldsymbol{\mathcal{M}})p(\boldsymbol{h}_T|\boldsymbol{\mathcal{M}},\pi)p(\boldsymbol{z}|\boldsymbol{\mathcal{M}})}\left[\log\frac{q_{\boldsymbol{\lambda}}(\boldsymbol{z}|f_{\boldsymbol{\phi}}(\boldsymbol{h}_T))}{p(\boldsymbol{z})}\right]$$

$$= \mathbb{E}_{p(\boldsymbol{\mathcal{M}})p(\boldsymbol{h}_T|\boldsymbol{\mathcal{M}},\pi)p(\boldsymbol{z}|\boldsymbol{\mathcal{M}})}\left[\log\frac{p(\boldsymbol{z}|\boldsymbol{h}_T)}{q_{\boldsymbol{\lambda}}(\boldsymbol{z}|f_{\boldsymbol{\phi}}(\boldsymbol{h}_T))}\right]$$

$$= \mathbb{E}_{p(\boldsymbol{\mathcal{M}},\boldsymbol{h}_T,\boldsymbol{z}|\pi)}\left[\log\frac{p(\boldsymbol{z}|\boldsymbol{h}_T)}{q_{\boldsymbol{\lambda}}(\boldsymbol{z}|f_{\boldsymbol{\phi}}(\boldsymbol{h}_T))}\right]$$

$$= \mathbb{E}_{p(\boldsymbol{h}_T,\boldsymbol{z}|\pi)}\left[\log\frac{p(\boldsymbol{z}|\boldsymbol{h}_T)}{q_{\boldsymbol{\lambda}}(\boldsymbol{z}|f_{\boldsymbol{\phi}}(\boldsymbol{h}_T))}\right]$$

$$= \mathbb{E}_{p(\boldsymbol{h}_T|\pi)p(\boldsymbol{z}|\boldsymbol{h}_T)}\left[\log\frac{p(\boldsymbol{z}|\boldsymbol{h}_T)}{q_{\boldsymbol{\lambda}}(\boldsymbol{z}|f_{\boldsymbol{\phi}}(\boldsymbol{h}_T))}\right]$$

$$= \mathbb{E}_{p(\boldsymbol{h}_T|\pi)}\left[D_{\mathrm{KL}}\Big(p(\boldsymbol{z}|\boldsymbol{h}_T) \,\|\, q_{\boldsymbol{\lambda}}(\boldsymbol{z}|f_{\boldsymbol{\phi}}(\boldsymbol{h}_T))\Big)\right] \tag{A4}$$

is an expectation of a Kullback–Leibler (KL) divergence, which is always non-negative. Hence, $\mathcal{I}_T(\pi) \geq \mathcal{I}_{T;L}(\pi; \boldsymbol{\lambda}, \boldsymbol{\phi})$ for any $\pi$, $\boldsymbol{\lambda}$, and $\boldsymbol{\phi}$. The bound is tight if and only if the KL divergence equals zero, which occurs when $q_{\boldsymbol{\lambda}}(\boldsymbol{z}|f_{\boldsymbol{\phi}}(\boldsymbol{h}_T)) = p(\boldsymbol{z}|\boldsymbol{h}_T)$ for all $\boldsymbol{z}$ and $\boldsymbol{h}_T$. □

## A.3 PRIOR-OMITTED EIG

We note that

$$\mathcal{I}_T(\pi) = \mathbb{E}_{p(\boldsymbol{\mathcal{M}})p(\boldsymbol{h}_T|\boldsymbol{\mathcal{M}},\pi)p(\boldsymbol{z}|\boldsymbol{\mathcal{M}})}\left[\log p(\boldsymbol{z}|\boldsymbol{h}_T)\right] - c$$
$$= \mathcal{R}_T(\pi) - c, \tag{A5}$$

where $c := \mathbb{E}_{p(\boldsymbol{\mathcal{M}})p(\boldsymbol{z}|\boldsymbol{\mathcal{M}})}[\log p(\boldsymbol{z})]$ is independent of $\pi$. Similarly,

$$\mathcal{I}_{T;L}(\pi; \boldsymbol{\lambda}, \boldsymbol{\phi}) = \mathbb{E}_{p(\boldsymbol{\mathcal{M}})p(\boldsymbol{h}_T|\boldsymbol{\mathcal{M}},\pi)p(\boldsymbol{z}|\boldsymbol{\mathcal{M}})}\left[\log q_{\boldsymbol{\lambda}}(\boldsymbol{z}|f_{\boldsymbol{\phi}}(\boldsymbol{h}_T))\right] - c$$
$$= \mathcal{R}_{T;L}(\pi; \boldsymbol{\lambda}, \boldsymbol{\phi}) - c. \tag{A6}$$

**Proposition A.2.** *For any policy $\pi$, variational parameter $\boldsymbol{\lambda}$, and embedding parameter $\boldsymbol{\phi}$, the prior-omitted EIG satisfies $\mathcal{R}_T(\pi) \geq \mathcal{R}_{T;L}(\pi; \boldsymbol{\lambda}, \boldsymbol{\phi})$. The bound is tight if and only if $p(\boldsymbol{z}|\boldsymbol{h}_T) = q_{\boldsymbol{\lambda}}(\boldsymbol{z}|f_{\boldsymbol{\phi}}(\boldsymbol{h}_T))$ for all $\boldsymbol{z}$ and $\boldsymbol{h}_T$.*

*Proof.* $\mathcal{R}_T(\pi) = \mathcal{I}_T(\pi) + c \geq \mathcal{I}_{T;L}(\pi; \boldsymbol{\lambda}, \boldsymbol{\phi}) + c = \mathcal{R}_{T;L}(\pi; \boldsymbol{\lambda}, \boldsymbol{\phi})$, making use of $\mathcal{I}_T(\pi) \geq \mathcal{I}_{T;L}(\pi; \boldsymbol{\lambda}, \boldsymbol{\phi})$ from Theorem 4.1. □

We adopt standard Monte Carlo to estimate $\mathcal{R}_{T;L}(\pi; \boldsymbol{\lambda}, \boldsymbol{\phi})$:

$$\mathcal{R}_{T;L}(\pi; \boldsymbol{\lambda}, \boldsymbol{\phi}) = \mathbb{E}_{p(\boldsymbol{\mathcal{M}})p(\boldsymbol{h}_T|\boldsymbol{\mathcal{M}},\pi)p(\boldsymbol{z}|\boldsymbol{\mathcal{M}})}\left[\log q_{\boldsymbol{\lambda}}(\boldsymbol{z}|f_{\boldsymbol{\phi}}(\boldsymbol{h}_T))\right]$$
$$\approx \frac{1}{N}\sum_{i=1}^{N}\log q_{\boldsymbol{\lambda}}(\boldsymbol{z}^i|f_{\boldsymbol{\phi}}(\boldsymbol{h}_T^i)), \tag{A7}$$

where $\boldsymbol{\mathcal{M}}^i \sim p(\boldsymbol{\mathcal{M}})$, $\boldsymbol{h}_T^i \sim p(\boldsymbol{h}_T|\boldsymbol{\mathcal{M}}^i, \pi)$, and $\boldsymbol{z}^i \sim p(\boldsymbol{z}|\boldsymbol{\mathcal{M}}^i)$.

We further propose a NMC estimator to estimate $\mathcal{R}_T(\pi)$:

$$
\begin{aligned}
\mathcal{R}_T(\pi) &= \mathbb{E}_{p(\mathcal{M})p(\boldsymbol{h}_T|\mathcal{M},\pi)p(\boldsymbol{z}|\mathcal{M})}\left[\log p(\boldsymbol{z}|\boldsymbol{h}_T)\right] \\
&= \mathbb{E}_{p(\mathcal{M})p(\boldsymbol{h}_T|\mathcal{M},\pi)p(\boldsymbol{z}|\mathcal{M})}\left[\log p(\boldsymbol{z},\boldsymbol{h}_T|\pi) - \log p(\boldsymbol{h}_T|\pi)\right] \\
&= \mathbb{E}_{p(\mathcal{M})p(\boldsymbol{h}_T|\mathcal{M},\pi)p(\boldsymbol{z}|\mathcal{M})}\left[\log \mathbb{E}_{p(\mathcal{M}')}[p(\boldsymbol{z},\boldsymbol{h}_T|\mathcal{M}',\pi)] - \log \mathbb{E}_{p(\mathcal{M}'')}[p(\boldsymbol{h}_T|\mathcal{M}'',\pi)]\right] \\
&\approx \frac{1}{N}\sum_{i=1}^{N}\left[\log\frac{1}{M_1}\sum_{j_1=1}^{M_1}p(\boldsymbol{z}^i,\boldsymbol{h}_T^i|\mathcal{M}^{j_1},\pi) - \log\frac{1}{M_2}\sum_{j_2=1}^{M_2}p(\boldsymbol{h}_T^i|\mathcal{M}^{j_2},\pi)\right],
\end{aligned}
\tag{A8}
$$

where $\mathcal{M}^i \sim p(\mathcal{M})$, $\boldsymbol{h}_T^i \sim p(\boldsymbol{h}_T|\mathcal{M}^i,\pi)$, $\boldsymbol{z}^i \sim p(\boldsymbol{z}|\mathcal{M}^i)$, and $\mathcal{M}^{j_1} \sim p(\mathcal{M}')$ and $\mathcal{M}^{j_2} \sim p(\mathcal{M}'')$. This NMC estimator is only used in Section 5.1 as a baseline comparison.

### A.4 NORMALIZING FLOWS

An NF is an invertible transformation that maps a target random variable $\boldsymbol{z}$ to a standard normal random variable $\boldsymbol{\eta}$, such that $\boldsymbol{z} = g(\boldsymbol{\eta})$ and $\boldsymbol{\eta} = f(\boldsymbol{z})$, where $f = g^{-1}$. The probability densities of $\boldsymbol{z}$ and $\boldsymbol{\eta}$ are related via the change-of-variables formula:

$$
p(\boldsymbol{z}) = p_{\boldsymbol{\eta}}(f(\boldsymbol{z}))\left|\det\frac{\partial f(\boldsymbol{z})}{\partial \boldsymbol{z}}\right|.
\tag{A9}
$$

Lt the transformation $g$ be expressed as a composition of $n \geq 1$ successive invertible functions: $\boldsymbol{z} = g(\boldsymbol{\eta}) = g_1 \circ g_2 \circ \ldots \circ g_n(\boldsymbol{\eta}) = g_1(g_2(\ldots(g_n(\boldsymbol{\eta}))\ldots))$. Then, the corresponding log-density of $\boldsymbol{z}$ becomes:

$$
\log p(\boldsymbol{z}) = \log p_{\boldsymbol{\eta}}(f_n \circ f_{n-1} \circ \ldots \circ f_1(\boldsymbol{z})) + \sum_{i=1}^{n}\log\left|\det\frac{\partial f_i \circ f_{i-1} \circ \ldots f_1(\boldsymbol{z})}{\partial \boldsymbol{z}}\right|,
\tag{A10}
$$

where $\boldsymbol{\eta} = f(\boldsymbol{z}) = f_n \circ f_{n-1} \circ \ldots \circ f_1(\boldsymbol{z})$ and $f_i = g_i^{-1}$. Through these successive transformations, NFs can model highly expressive and flexible densities for the target variable $\boldsymbol{z}$ Dinh et al. (2016).

To approximate the QoI posterior $q_{\boldsymbol{\lambda}}(\boldsymbol{z}|f_{\boldsymbol{\phi}}(\boldsymbol{h}_T))$, we employ NFs composed of successive coupling layers. Each coupling layer partitions $\boldsymbol{z}$ into two similarly sized subsets, $\boldsymbol{z} = [\boldsymbol{z}_1, \boldsymbol{z}_2]^\top$, with dimensions $n_{\boldsymbol{z}_1}$ and $n_{\boldsymbol{z}_2}$, respectively. The coupling transformations are defined as:

$$
\begin{aligned}
f_1(\boldsymbol{z}) &= \begin{pmatrix} \boldsymbol{z}_1 \\ \tilde{\boldsymbol{z}}_2 := \boldsymbol{z}_2 \odot \exp(s_1(\boldsymbol{z}_1)) + t_1(\boldsymbol{z}_1) \end{pmatrix} \\
f_2(f_1(\boldsymbol{z})) &= \begin{pmatrix} \tilde{\boldsymbol{z}}_1 := \boldsymbol{z}_1 \odot \exp(s_2(\tilde{\boldsymbol{z}}_2)) + t_2(\tilde{\boldsymbol{z}}_2) \\ \tilde{\boldsymbol{z}}_2 \end{pmatrix},
\end{aligned}
\tag{A11}
$$

where $s_1, t_1 : \mathbb{R}^{n_{\boldsymbol{z}_1}} \mapsto \mathbb{R}^{n_{\boldsymbol{z}_2}}$ and $s_2, t_2 : \mathbb{R}^{n_{\boldsymbol{z}_2}} \mapsto \mathbb{R}^{n_{\boldsymbol{z}_1}}$ are flexible mappings (e.g., neural networks), and $\odot$ denotes the element-wise product. The Jacobian of the transformation $f_1$ is given by:

$$
\begin{bmatrix} \mathbb{I}_d & 0 \\ \frac{\partial f_1(\boldsymbol{z})}{\partial \boldsymbol{z}_2} & \mathrm{diag}(\exp(s_1(\boldsymbol{z}_1))) \end{bmatrix},
$$

which is lower-triangular with determinant $\exp(\sum_{j=1}^{n_{\boldsymbol{z}_2}} s_1(\boldsymbol{z}_1)_j)$. Similarly, the Jacobian of $f_2$ is upper-triangular with determinant $\exp(\sum_{j=1}^{n_{\boldsymbol{z}_1}} s_2(\tilde{\boldsymbol{z}}_2)_j)$. Multiple coupling transformations ($n_{\text{trans}}$) from equation A11 can be composed sequentially to increase the expressive power of the overall transformation. To capture the dependencies of the intervention history $\boldsymbol{h}_T$, we additionally condition the mappings $s(\cdot)$ and $t(\cdot)$ on the embedding $f_{\boldsymbol{\phi}}(\boldsymbol{h}_T)$.

## B DETAILED RELATED WORK

Our work on GO-CBED builds upon several related lines of research.

**Causal Bayesian Experimental Design**   Experimental design for causal discovery within a BOED framework was initially explored by Murphy (2001) and Tong & Koller (2001) for discrete variables with single-target acquisition. Subsequent research extended this approach to continuous variables within BOED (Agrawal et al., 2019; von Kügelgen et al., 2019; Toth et al., 2022; Cho et al., 2016) and alternative frameworks (Kocaoglu et al., 2017a; Gamella & Heinze-Deml, 2020; Ghassami et al., 2018; Olko et al., 2024). Notable non-BOED methods include strategies for cyclic structures (Mokhtarian et al., 2022) and latent variables (Kocaoglu et al., 2017b). Within BOED, Tigas et al. (2022) proposed selecting single target-state pairs via stochastic batch acquisition, later extending this to gradient-based optimization to multiple target-state pairs (Tigas et al., 2023). Sussex et al. (2021) introduced a greedy method for selecting multi-target experiments without specifying intervention states. More recently, Annadani et al. (2024) proposed adaptive sequential experimental design for causal structure learning, although their objective—minimizing graph prediction error—is different from traditional BOED. Gao et al. (2024) developed a reinforcement learning method for sequential experimental design using Prior Contrastive Estimation (Foster et al., 2021) as a reward function; however, their approach relies on initial observational data and is computationally intensive. In contrast, our method uses direct policy optimization with differentiable rewards, enabling more efficient training without needing initial observational data.

**Bayesian Causal Discovery**   Causal discovery has been extensively studied in machine learning and statistics (Glymour et al., 2019; Heinze-Deml et al., 2018; Peters et al., 2017; Vowels et al., 2022). Traditional causal discovery methods typically infer a single causal graph from observational data (Brouillard et al., 2020; Hauser & Bühlmann, 2012; Lippe et al., 2021; Perry et al., 2022; Peters et al., 2016; Heinze-Deml et al., 2018). In contrast, Bayesian causal discovery (Friedman & Koller, 2003; Heckerman et al., 2006; Tong & Koller, 2001) seeks to infer a posterior distribution over SCMs. Recent work (Cundy et al., 2021; Lorch et al., 2021; Annadani et al., 2021) has introduced variational approximations of the DAG posterior, enabling representation of uncertainty by a full distribution rather than a point estimate. Addressing the discrete nature of DAGs—which prevents straightforward gradient-based optimization—Lorch et al. (2021) used Stein variational gradient descent (SVGD) (Liu & Wang, 2016) in a continuous latent embedding space, enabling efficient Bayesian inference over DAG structures.

**Goal-Oriented and Decision-Theoretic BOED**   Goal-oriented BOED extends classical optimal design principles—such as L-, $D_A$-, I-, V-, and G-optimality (Atkinson et al., 2007)—by shifting the objective from general parameter estimation to directly maximizing utility for specific, downstream QoIs, a concept first formulated by Bernardo (1979). Modern work has focused on the computational challenges of this paradigm, developing scalable approximations for high-dimensional QoIs (Attia et al., 2018; Wu et al., 2021) and leveraging advanced sampling or likelihood-free methods for complex nonlinear scenarios (Zhong et al., 2024; Chakraborty et al., 2024). Within this landscape, a prominent direction is decision-theoretic BED, which optimizes experiments for downstream task performance. For instance, Huang et al. (2024) use an amortized transformer policy to maximize a Decision Utility Gain, while Filstroff et al. (2021) introduce an active learning criterion that directly maximizes the expected information gain over the posterior of the optimal decision, framing utility in terms of outcome predictions $y$ rather than parameter inference over $\theta$. Similarly, in the causal domain, Kim et al. (2024) demonstrate that targeted approaches can be more efficient when identifying specific causal features rather than complete models, showing how to learn causal parents of target variables from observational data without recovering full graphs. Closer to our approach, other methods use information-theoretic objectives for specific goals. A key example is Bayesian Algorithm Execution (BAX), which maximizes information gain to estimate properties of black-box functions (Neiswanger et al., 2021). However, despite their philosophical alignment with our work, these powerful frameworks are not designed for the unique challenges of causal experimental design. Their core limitations are twofold: they typically operate on static functions with standard uncertainty, not on Structural Causal Models (SCMs) with their hybrid uncertainty space that combines discrete graphs and continuous, graph-dependent mechanisms; and they query a fixed data-generating process, unable to accommodate an interventional action space where experiments surgically alter the probabilistic model itself to answer causal questions.

**Non-Myopic Sequential BOED**   Non-myopic sequential BOED addresses the limtiations of greedy, single-step experimental strategies by planning optimal sequences of interventions. Such methods

have been broadly explored in various general settings (Foster et al., 2021; Ivanova et al., 2021; Blau et al., 2022; Shen & Huan, 2023; Blau et al., 2023), including goal-oriented extensions such as vsOED (Shen et al., 2025). Within causal BOED specifically, non-myopic approaches have predominantly focused on causal discovery tasks aimed at learning the graph structures (Annadani et al., 2024; Gao et al., 2024). Although active learning methods targeting specific causal reasoning queries have also been proposed (Toth et al., 2022), these typically employ myopic (single-step) intervention designs, thus limiting their ability to strategically plan for long-term gains.

## C  EXPERIMENT DETAILS

### C.1  HYPERPARAMETER SETTINGS FOR POLICY AND POSTERIOR NETWORKS

The input to the policy network has shape ($n = n_{\text{int}} \times T$, $d$, 2), where the last dimension encodes the intervention data and binary intervention masks. The policy network architecture (see Figure 7) proceeds as follows:

1. The input is passed through a fully connected layer, transforming it to the shape ($n_{\text{int}} \times T$, $d$, $n_{\text{embedding}}$).

2. The embedded representation is processed through $L$ stacked Transformer layers. Each layer includes:
   - Two multi-head self-attention sublayers, each preceded by layer normalization and followed by dropout.
   - A feedforward fully-connected (FFN) sublayer, also preceded by layer normalization and followed by dropout.

   Residual connections are applied after each sublayer. This output retains the shape ($n_{\text{int}} \times T$, $d$, $n_{\text{embedding}}$).

3. A max-pooling operation is applied across the $n_{\text{int}} \times T$ dimension, yielding a compressed representation of shape ($d$, $n_{\text{embedding}}$).

4. The pooled representation is passed through:
   - A target prediction layer, followed by a Gumbel-softmax transformation with temperature $\tau$, producing a discrete intervention target vector.
   - A separate value layer, with final outputs scaled to fall within a specific range $\min_{val}$ and $\max_{val}$.

The detailed implementation setup is provided in Table 1. The "step" associated with $\tau$ refers to the current training step, and the values of $T$, $n_{\text{step}}$, and $n_{\text{envs}}$ per training step are kept to be the same as those used for training the posterior networks (see below).

For the posterior networks, the initial input has shape ($n_{\text{envs}}$, $n_{\text{int}} \times T$, $d$, 2), representing full trajectories. The processing steps follows the same as those of the policy network up to step 3, resulting in a max-pooled output of shape ($n_{\text{envs}}$, $d$, $n_{\text{embedding}}$). Specific to the causal discovery case, starting from step 4:

4. The pooled representation is processed as follows:
   - Two independent linear transformations are applied to produce vectors $\boldsymbol{u}$ and $\boldsymbol{v}$, each of shape ($n_{\text{envs}}$, $d$, $n_{\text{out}}$).
   - Both $\boldsymbol{u}$ and $\boldsymbol{v}$ are normalized using their $\ell_2$-norm along the last dimension.

5. Pairwise edge logits are computed:
   - A dot product between every pair of variables $\boldsymbol{u}_i$ and $\boldsymbol{v}_j$, resulting in a tensor of shape ($n_{\text{envs}}$, $d$, $d$).
   - The logits are scaled by a learnable temperature parameter "temp" via the operation $\text{logit}_{ij} \times \exp(\text{temp})$, which is then added element-wise with a learnable term, "bias".

The detailed implementation setup is provided in Table 2.

Specific to the causal reasoning case, starting from step 4:

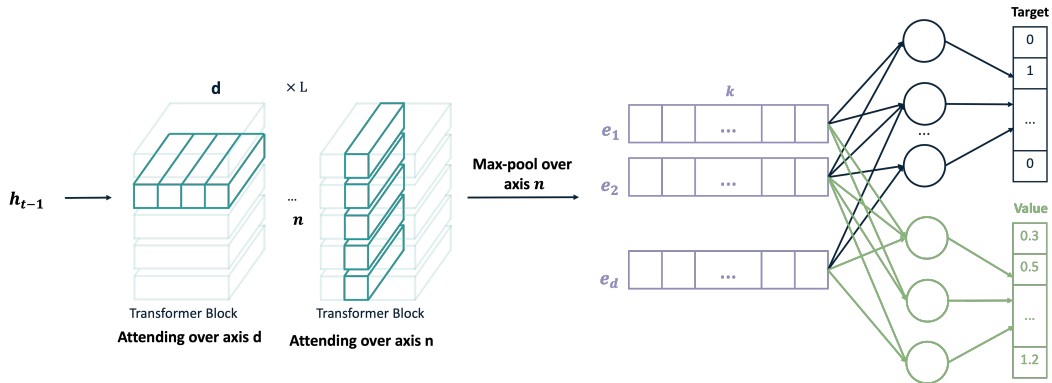

Figure 7: Policy network architecture. The model takes as input a three-dimensional tensor of shape $n \times d \times 2$, where $n = n_{\text{int}} \times T$. It is permutation-invariant along the $n$-axis and permutation-equivariant along the $d$-axis. Each of the $L$ layers first applies self-attention across the $d$-axis, followed by attention across the $n$-axis, with shared parameters across the non-attended axis.

Table 1: Hyperparameter settings for the policy network.

| Hyperparameter | Value |
|---|---|
| Embedding dimension $n_{\text{embedding}}$ | 32 |
| Number of transformer layers ($L$) | 4 |
| Key size in self-attention | 16 |
| Number of attention heads | 8 |
| FFN dimensions | $(n_{\text{embedding}}, 4 \times n_{\text{embedding}}, n_{\text{embedding}})$ |
| Activation | ReLU |
| Dropout rate | 0.05 |
| $\max_{val}$ | 10 |
| $\min_{val}$ | $-10$ |
| $\tau$ | $\min(5 \times 0.9995^{\text{step}}, 0.1)$ |
| Initial learning rate | $5 \times 10^{-4}$ or $10^{-4}$ |
| Scheduler | ExponentialLR with $\gamma = 0.8$, step every 1000 training steps |
| $T$ | 10 when $d = 10, 20$ |
| | ~~15000~~ 15 when $d = 30$ |
| $n_{\text{step}}$ | 10000 when $d = 10$ |
| | 15000 when $d = 20, 30$ |
| $n_{\text{env}}$ per training step | 10 |

Table 2: Hyperparameter settings for the posterior network in the causal discovery case.

| Hyperparameter | Value |
|---|---|
| Embedding dimension $n_{\text{embedding}}$ | 128 |
| Number of transformer layers ($L$) | 8 |
| Key size in self-attention | 64 |
| Number of attention heads | 8 |
| FFN dimensions | $(n_{\text{embedding}}, 4 \times n_{\text{embedding}}, n_{\text{embedding}})$ |
| Activation | ReLU |
| Dropout rate | 0.05 |
| Bias | $-3$ |
| Temp | 2 |
| Initial learning rate | $10^{-4}$ |
| Scheduler | ExponentialLR with $\gamma = 0.8$, step every 1000 training steps |

4. The pooled representation is flattened to shape $(n_{\text{envs}}, d \times n_{\text{embedding}})$ and passed into the $s(\cdot)$ and $t(\cdot)$ networks, with $n_{\text{trans}}$ transformations in total. The final output has shape $(n_{\text{envs}}, n_{\boldsymbol{z}})$.

The detailed implementation setup is provided in Table 3.

Table 3: Hyperparameter settings for the posterior network in the causal reasoning case.

| Hyperparameter | Value |
|---|---|
| Embedding dimension $n_{\text{embedding}}$ | 16 |
| Number of transformer layers ($L$) | 8 |
| Key size in self-attention | 16 |
| Number of attention heads | 8 |
| FFN dimensions | $(n_{\text{embedding}}, 4 \times n_{\text{embedding}}, n_{\text{embedding}})$ |
| Activation | ReLU |
| Dropout rate | 0.05 |
| $n_{\text{trans}}$ | 4 |
| $s(\cdot)$ and $t(\cdot)$ dimensions | (256, 256, 256) |
| Initial learning rate | $5 \times 10^{-4}$ or $10^{-3}$ |
| Scheduler | ExponentialLR with $\gamma = 0.8$, step every 1000 training steps |

### C.2 EXAMPLE IN SECTION 5.1

In this example, we consider a fixed causal graph structure with Gaussian priors on parameters:

$$\theta_{12} \sim \mathcal{N}(0.1, 1), \quad \theta_{23} \sim \mathcal{N}(1, 0.2^2), \quad \theta_{35} \sim \mathcal{N}(0.2, 0.5^2), \quad \theta_{45} \sim \mathcal{N}(-0.5, 0.5^2),$$
$$\theta_{13} \sim \mathcal{N}(-0.2, 0.5^2), \quad \theta_{24} \sim \mathcal{N}(0.3, 0.3^2), \quad \theta_{36} \sim \mathcal{N}(0, 0.5^2), \quad \theta_{56} \sim \mathcal{N}(0, 0.5^2),$$
$$\theta_{14} \sim \mathcal{N}(-0.5, 0.3^2),$$

with observation model $X_i = \boldsymbol{\theta}_i^\top \boldsymbol{X}_{\text{pa}(i)} + \epsilon_i$ where $\boldsymbol{\theta}_i = [\theta_{ij}]^\top, j \in \text{pa}(i)$, and additive Gaussian noise $\epsilon_i \sim \mathcal{N}(0, \sigma_i^2)$ with standard deviations $\boldsymbol{\sigma} = \{0.2, 0.2, 0.2, 0.2, 0.3, 0.3\}$. This linear-Gaussian setup enables analytical posterior computations and efficient estimation of the shifted EIG $\mathcal{R}_T(\pi; \phi)$.

To strengthen the EIG-based comparison, Figure 8 evaluates the Wasserstein distance between posteriors for trajectories generated by different policies. The results show consistent trends: policies achieving higher EIG also produce posteriors closer to the ground truth in Wasserstein distance.

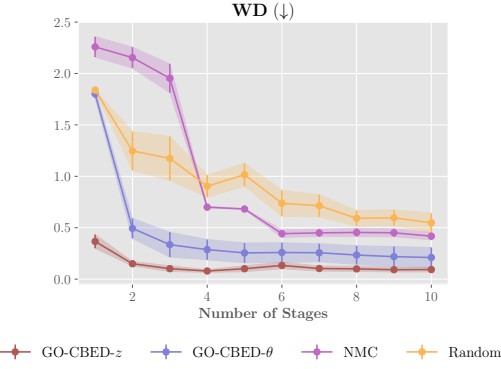

Figure 8: Performance comparison of policies trained for $T = 10$ on causal queries. GO-CBED-z consistently outperforms the other approaches.

Figure 9 additionally provides a qualitative assessment of the posterior approximation $q_{\boldsymbol{\lambda}}(\boldsymbol{z}|f_{\phi}(\boldsymbol{h}_T))$ achieved by NFs. The NF-based approximations closely align with the true posterior predictive distributions $p(\boldsymbol{z}|h_T)$ across these two examples, demonstrating a high-quality posterior approximation.

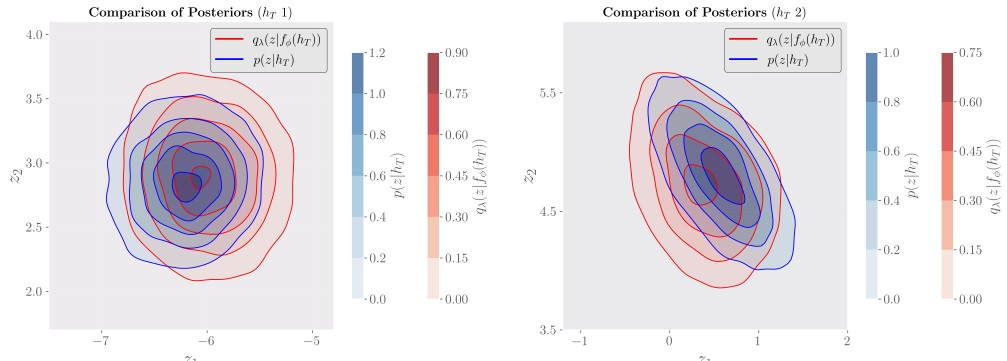

Figure 9: Comparison between the true posterior predictive distribution $p(\boldsymbol{z}|\boldsymbol{h}_T)$ and the variational approximation $q_{\boldsymbol{\lambda}}(\boldsymbol{z}|f_{\boldsymbol{\phi}}(\boldsymbol{h}_T))$ for two simulated trajectories. The approximate posterior closely aligns with the true posterior.

### C.3 EXAMPLES IN SECTIONS 5.2 AND 5.3

In the synthetic experiments, we consider Erdös–Rényi (ER) and Scale-free (SF) random graphs as priors over graph structures. For semi-synthetic experiments, we utilize gene regulatory networks derived from the DREAM benchmarks Greenfield et al. (2010), which reflect realistic biological scenarios.

#### C.3.1 PRIORS OVER GRAPH STRUCTURES

**Erdös–Rényi**   In the ER model, each potential edge between node pairs is included independently with a fixed probability $p$. Given $n$ nodes, the resulting random undirected graph has a number of edges that follows a binomial distribution, with the expected number of edges equal to $p \times \binom{n}{2}$. Following Lorch et al. (2022), we scale $p$ such that the expected number of edges is $\mathcal{O}(d)$, where $d$ is the desired average degree. To obtain a DAG, we first retain only the lower triangular portion of the adjacency matrix and then apply a random permutation to the node indices to break symmetry.

**Scale-free**   SF graphs exhibit a power-law degree distribution, where the probability that a node has degree $k$ is proportional to $k^{-\gamma}$, with an exponent $\gamma > 1$ (Barabási & Albert, 1999). Consequently, a small subset of nodes ("hubs") has a very high number of connections, while most nodes have relatively few. Such structures are commonly observed in biological and social networks. We generate SF graphs using the Barabási–Albert preferential attachment model implemented in NetworkX, which iteratively adds nodes by connecting them preferentially to existing high-degree nodes.

**Realistic Gene Regulatory Networks**   For semi-synthetic scenarios, we employ networks from the DREAM benchmarks Greenfield et al. (2010), widely used for evaluating computational approaches to reverse-engineering biological systems. DREAM datasets provide realistic simulations of gene regulatory and protein signaling networks generated by GeneNetWeaver v3.12. Specifically, our experiments focus on two DREAM subnetworks—the *E. coli* and Yeast networks—following the setup described in Tigas et al. (2023).

#### C.3.2 MECHANISMS

**Linear Model**   In the linear setting, each variable $X_i$ is modeled as a linear function of its parent variables $\boldsymbol{X}_{\text{pa}(i)}$ according to

$$X_i = \boldsymbol{\theta}_i^\top \boldsymbol{X}_{\text{pa}(i)} + b_i + \epsilon_i, \tag{A12}$$

where $\epsilon_i \sim \mathcal{N}(0, \sigma^2)$ with fixed variance $\sigma^2 = 0.1$. The parameters have priors $\boldsymbol{\theta}_i \sim \mathcal{N}(0, 2)$ and $b_i \sim \mathcal{U}(-1, 1)$.

**Nonlinear Model** For the nonlinear setting, the functional relationship between each child and its parent is modeled using a feedforward neural network with two hidden layers, each containing 8 ReLU-activated neurons. All weights and biases have standard normal priors.

For the causal reasoning experiments shown in Figure 3, the query QoIs for the four panels are: $z = \{X_6, X_8 \mid \text{do}(X_2 \sim \mathcal{N}(5, 2^2))\}; \{X_0, X_5 \mid \text{do}(X_6 \sim \mathcal{N}(3, 1))\}; \{X_3, X_5 \mid \text{do}(X_5 \sim \mathcal{N}(6, 0.5^2))\};$ and $\{X_3, X_4 \mid \text{do}(X_9 \sim \mathcal{N}(4, 1))\}$. For the *E. coli* case in Figure 4, the QoI is $z = \{X_6, X_8 \mid \text{do}(X_7 \sim \mathcal{N}(4, 2^2))\}$.

**Comparison to Discovery-Oriented Policy** To contextualize the performance of GO-CBED-$z$, we also include the performance of the structure-learning-oriented policy $\pi_G^*$, evaluated on $\mathcal{R}_{T;L}(\pi_G^*)$, as shown in in Figures 10 and 11. While GO-CBED-$G$ is effective for causal discovery, it is consistently outperformed by GO-CBED-$z$ when the objective is to estimate specific causal inquiries, as seen by comparing to Figures 3 and 4.

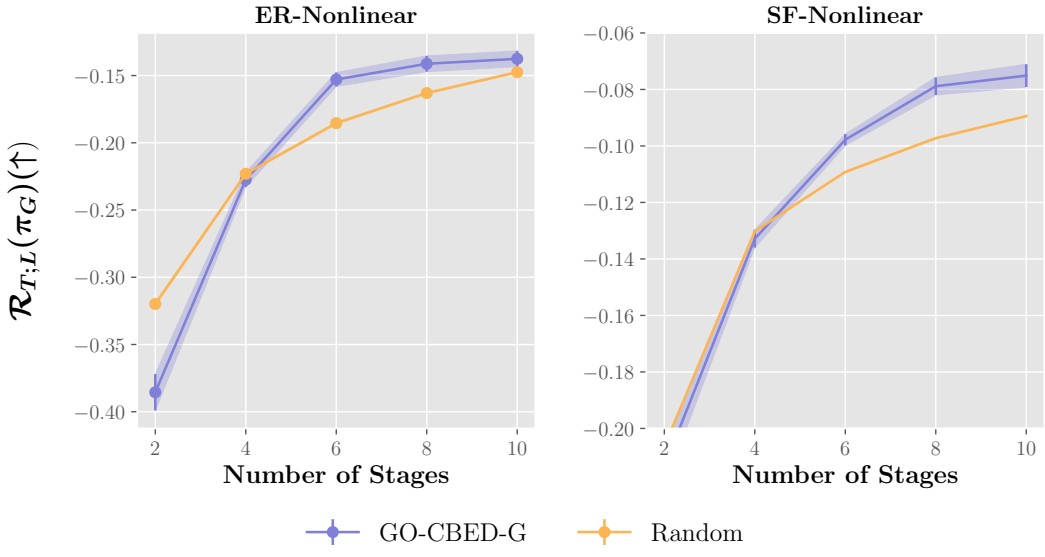

Figure 10: Evaluation of policies on ER and SF graphs with nonlinear causal mechanisms. The $\pi_G^*$ demonstrates better performance in identifying the underlying causal graph on both settings.

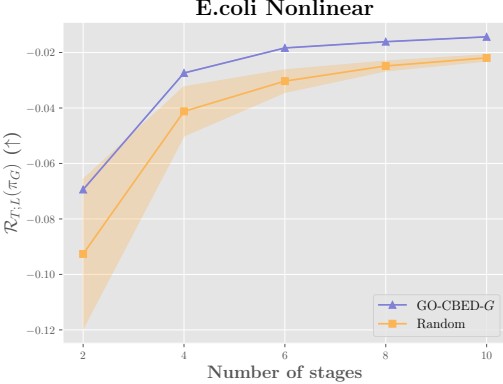

Figure 11: Evaluation of policies on *E. coli* graphs with nonlinear causal mechanisms. The $\pi_G^*$ demonstrates strong performance in accurately identifying the underlying causal graph.

### C.4 QUERIES FOR CAUSAL REASONING

Please see Table 4 for the ground truth graphs and corresponding target queries used in Section 5.2.

| Case | Graph Structure (Figure) |
|------|--------------------------|
| Erdős–Rényi | Figure 12 |
| Scale-Free | Figure 13 |
| GRN | Figure 16 |

Table 4: Ground truth graph and target quries used in causal reasoning tasks.

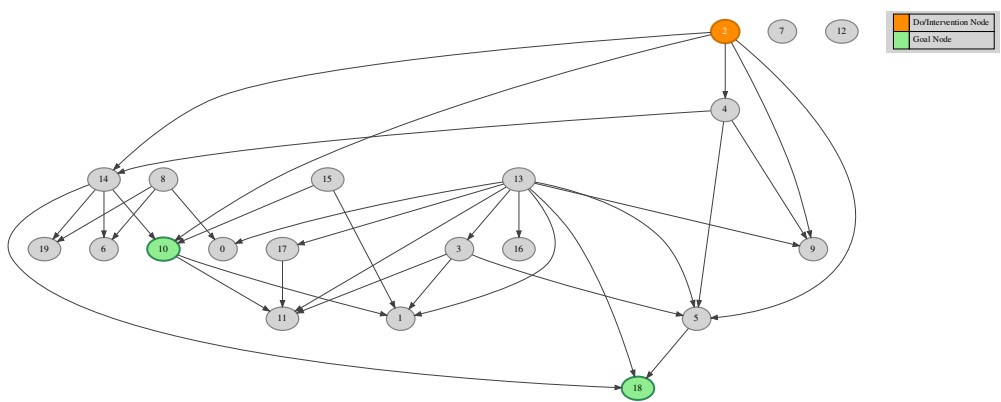

Figure 12: Ground truth ER graph used in Section 5.2.

## C.5 INCORPORATING EXISTING OBSERVATIONAL DATA INTO THE PRIOR

In practical settings, it is common to have access to a set of observational data $\mathcal{D}$ prior to designing interventions. This data can be used to update the prior into a posterior, which then serves as an informative prior for the subsequent experimental design. We infer both the posterior over the graph structure $p(G|\mathcal{D})$ and the parameters $p(\boldsymbol{\theta}|\mathcal{D}, G)$ in two stages.

First, since the realized data may not be available during posterior construction, we treat $\mathcal{D}$ as a random variable. We infer the graph structure using the approach of Lorch et al. (2022), and train an amortized variational posterior $q_{\boldsymbol{\lambda}}(f_{\boldsymbol{\phi}}(\mathcal{D}))$, which generalizes across potential realizations of $\mathcal{D}$, by minimizing

$$\mathbb{E}_{p(\mathcal{D})}\left[D_{\mathrm{KL}}\left(p(G|\mathcal{D}) \,||\, q_{\boldsymbol{\lambda}}(f_{\boldsymbol{\phi}}(\mathcal{D}))\right)\right] \tag{A13}$$

with respect to variational parameters $\boldsymbol{\lambda}$ and $\boldsymbol{\phi}$. The approximate posterior $q_{\boldsymbol{\lambda}}$ is modeled as a product of independent Bernoulli distributions over potential edges. Once a specific realization $\mathcal{D}^*$ becomes available, the posterior is instantiated via substitution as $q_{\boldsymbol{\lambda}}(f_{\boldsymbol{\phi}}(\mathcal{D}^*))$. Samples from this distribution are drawn from the Bernoulli marginals and retaining only acyclic graphs to ensure valid DAGs.

Second, to perform inference over the parameters $\boldsymbol{\theta}$, we exploit the conditional independence structure of the posterior:

$$p(\boldsymbol{\theta}|\mathcal{D}, G) = \prod_j p(\boldsymbol{\theta}_j|\mathcal{D}_{\mathrm{pa}(j)}, \mathcal{D}_j, G). \tag{A14}$$

This factorization enables efficient sampling of $\boldsymbol{\theta}$ by decomposing the joint posterior into node-wise conditionals. For linear models, we sample directly from the posterior $p(\boldsymbol{\theta}_j|G, \mathcal{D})$ using Markov chain Monte Carlo. For nonlinear models, we apply Pyro's Stochastic Variational Inference (SVI) Bingham et al. (2019) to learn a mean-field Gaussian approximation to the posterior.

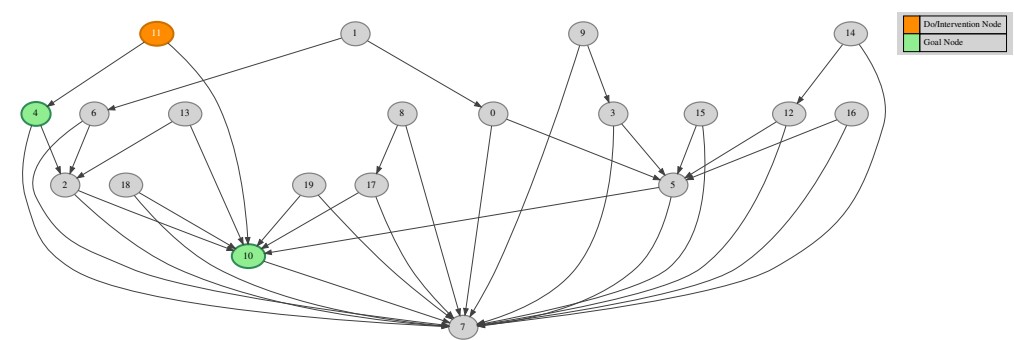

Figure 13: Ground truth Scale-Free graph used in Section 5.2.

# D    ADDITIONAL EXPERIMENTS

## D.1    HIGHER BIAS IN THE NMC ESTIMATOR

We provide a qualitative comparison between the NMC estimator and the GO-CBED approach. The experiment follows the setup in Figure 14, which assumes a fixed graph with $T = 1$ and additive Gaussian noise $\epsilon_i \sim \mathcal{N}(0, 0.3^2)$ for all observations. Interventions are uniformly selected in integers from $-5$ to $5$, and the $\mathcal{R}_T$ or $\mathcal{R}_{T;L}$ is evaluated using the NMC and GO-CBED estimators and presented in Figure 15. Since there is no policy optimization in this setting, GO-CBED reduces to training a variational posterior network using NFs.

The NMC estimator uses an outer loop size of $5,000$ samples, and the inner loop sample size is indicated in the parenthesis in the legend of Figure 15. This sample size is also used as the training sample size for GO-CBED. Despite using significantly fewer samples, GO-CBED consistently identifies the optimal EIG near the boundary of the design space. This finding reinforces our observation from Section 5.1: variational approximations via GO-CBED can offer more efficient and reliable EIG estimation compared to NMC, especially in causal inference tasks involving large graphs or high-dimensional parameter spaces, where traditional sampling becomes computationally and memory intensive.

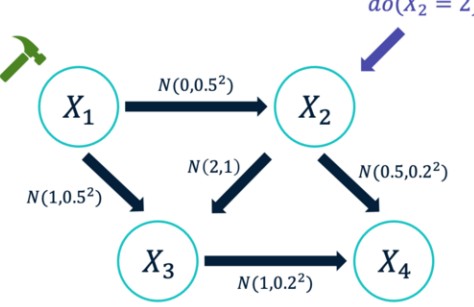

Figure 14: Evaluation of interventions on node 1 using integers from $-5$ to $5$, with the causal query defined as $z = \{X_3, X_4 \mid \mathrm{do}(X_2 = 2)\}$.

## D.2    CAUSAL REASONING ON DIVERSE REALISTIC GRAPH STRUCTURES

In Section 5.2, we presented causal reasoning results using the *E. coli* gene regulatory network. Here, we extend the analysis to include additional tasks based on both *E. coli* and Yeast gene regulatory networks, each incorporating nonlinear mechanisms. These supplementary experiments further demonstrate the robustness and generality of GO-CBED across a range of causal graph structures and varying complexities of intervention-target relationships (see Figure 16).

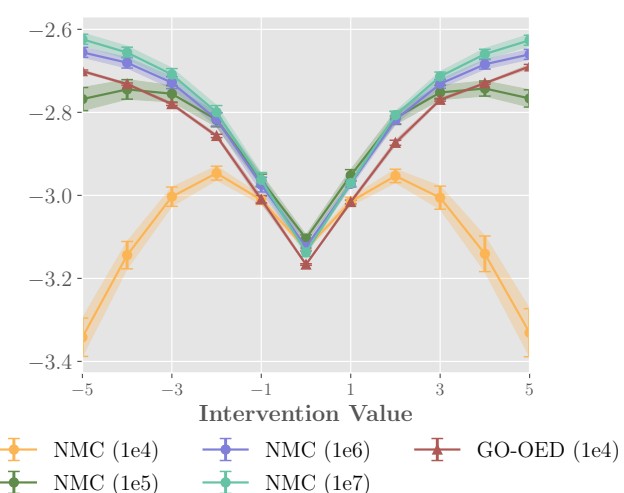

Figure 15: Prior-omitted EIG lower bound estimates, with parenthesis values denoting the inner loop sample size for NMC and training sample size for GO-CBED. Shaded regions represent $\pm 1$ standard error across 4 random seeds.

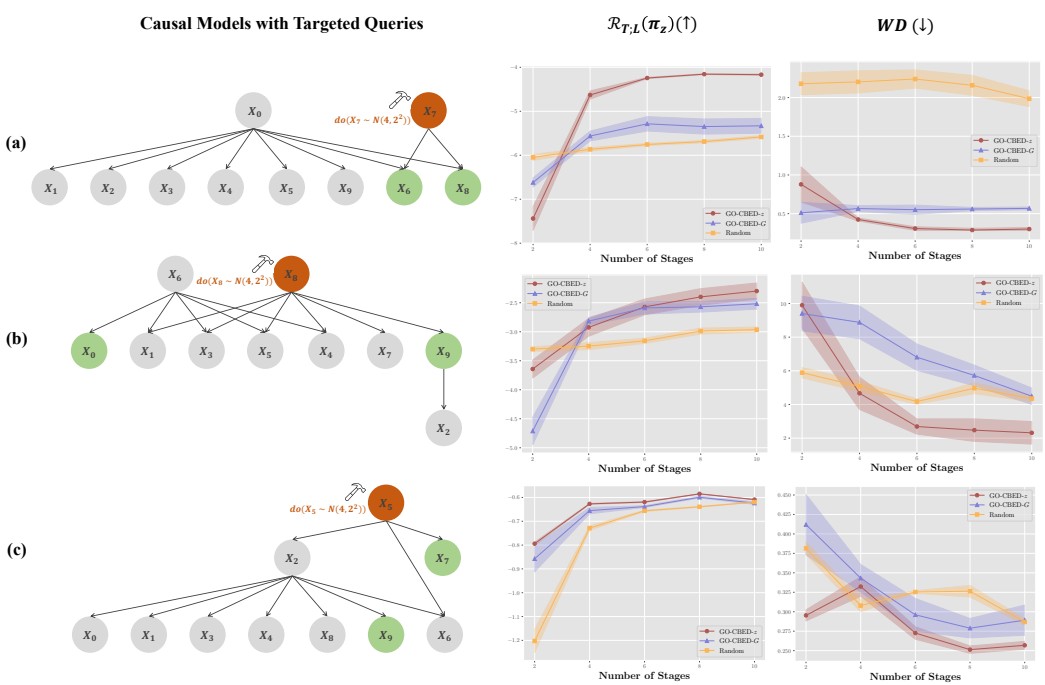

Figure 16: Additional causal reasoning experiments on nonlinear gene regulatory networks. **(a)** *E. coli* network in Section 5.2: intervention on node 7 targeting nodes 6 and 8. **(b)** Yeast network: intervention on node 8 targeting nodes 0 and 9. **(c)** *E. coli* network: intervention on node 5 targeting nodes 7 and 9. GO-CBED consistently outperforms baseline methods, with performance gains varying based on the structural complexity of the intervention-target relationships. Shaded regions represent $\pm 1$ standard error over 4 random seeds.

Specifically, Figure 16(a) corresponds to the main result in the paper. Figures 16)(b) and (c) illustrate additional scenarios, highlighting GO-CBED's consistent advantages across diverse topologies. In Figure 16)(b), GO-CBED achieves higher the EIG compared to baselines. This improvement likely stems from the complex paths linking intervention nodes to targets, where goal-oriented strategies more effectively exploit structural dependencies.

In contrast, Figure 16(c) shows a reduced performance gap. This is likely due to node $X_5$ being highly informative for both causal discovery and targeted queries, aligning the objective of structure learning and query-specific inference. Notably, the random policy also performs competitively in this setting, likely benefiting from the high-quality variational posterior achievable even under random interventions. We leave a deeper investigation of this phenomenon as an interesting direction for future work.

### D.3 EXTENDED EVALUATION ON CAUSAL DISCOVERY TASKS

We further evaluate in nonlinear settings: synthetic SCMs with ER and SF graph priors (Figures 17 and 18), and semi-synthetic *E. coli* and Yeast gene-regulatory networks (Figures 19 and 20). For nonlinear SCMs, we benchmark against Random and SoftCBED only, as other baselines were not validated in this regime in their original work. Across all the metrics, GO-CBED performs better or comparatively compared to these baselines, though the margins are smaller than in the linear case, reflecting the added difficulty of recovering full structures with nonlinear mechanisms. These findings reinforce the motivation for goal-oriented BOED: when the objective is to answer specific causal queries rather than reconstruct the entire model, targeted policies provide greater efficiency.

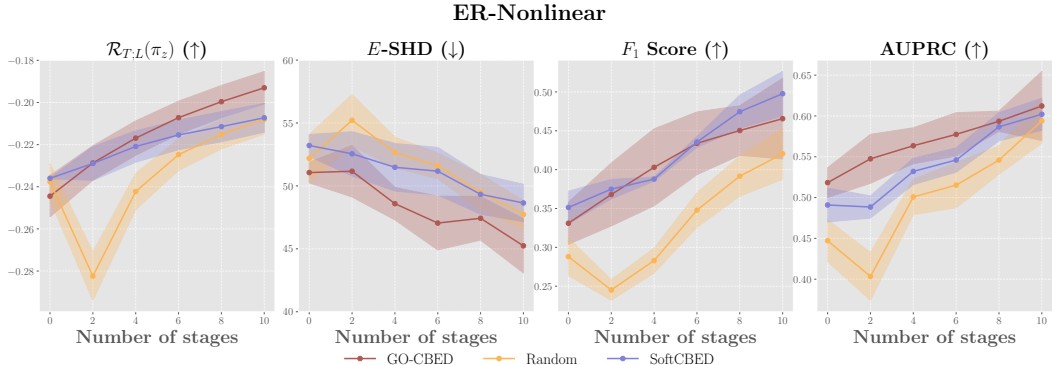

Figure 17: Causal discovery performance on nonlinear SCMs with Erdős-Rényi (ER) prior. GO-CBED performs better or comparatively in terms of uncertainty reduction ($\mathcal{R}_{T;L}$), structural recovery ($\mathbb{E}$-SHD), and structural accuracy ($F_1$-score and AUPRC) compared to all baselines. Shaded regions indicate $\pm 1$ standard error across 10 random seeds.

### D.4 PERFORMANCE COMPARISON WITH ORIGINAL POSTERIOR INFERENCE METHODS

In the main paper, we establish a fair comparison among policies by evaluating them using their respectively trained variational posteriors, thereby isolating the impact of policy optimization. However, a key advantage of GO-CBED is its joint training of both the policy and posterior networks. Specifically, the posterior networks trained in GO-CBED can themselves serve as a highly efficient inference tool, even independent of the policy.

Here, we provide additional results in which each baseline method is evaluated using its original posterior inference procedure, as proposed in its respective paper. Specifically, CAASL (Annadani et al., 2024) uses a fixed pre-trained AVICI posterior (Lorch et al., 2021); DiffCBED (Tigas et al., 2023) and SoftCBED (Tigas et al., 2022) rely on DAG-Bootstrap (Friedman et al., 2013; Hauser & Bühlmann, 2012) for linear SCMs and DiBS (Lorch et al., 2021) for nonlinear SCMs. For brevity, we report $\mathbb{E}$-SHD and $F_1$ as representative structure- and edge-level metrics.

Figures 21 and 22 present results on synthetic and semi-synthetic SCMs with both linear and nonlinear mechanisms. GO-CBED consistently outperforms baselines across both the $\mathbb{E}$-SHD and $F_1$ score

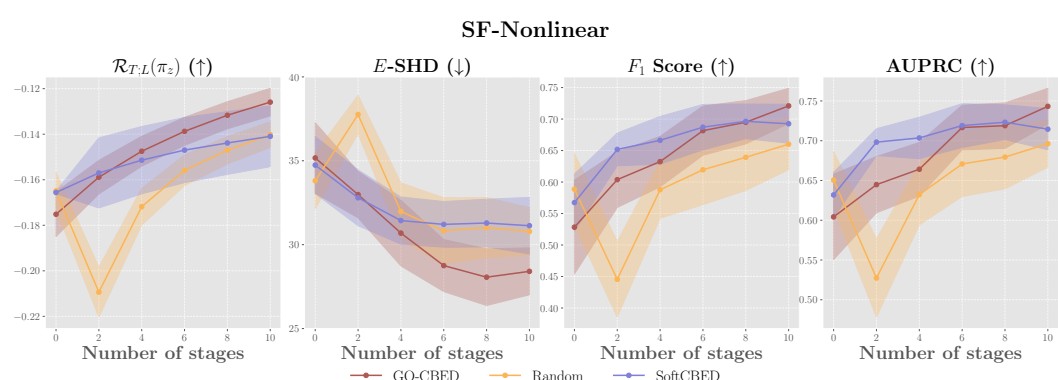

Figure 18: Causal discovery performance on nonlinear SCMs with Scale-free (SF) prior. GO-CBED performs better or comparatively in terms of uncertainty reduction ($\mathcal{R}_{T;L}$), structural recovery ($\mathbb{E}$-SHD), and structural accuracy ($F_1$-score and AUPRC) compared to all baselines. Shaded regions indicate $\pm 1$ standard error across 10 random seeds.

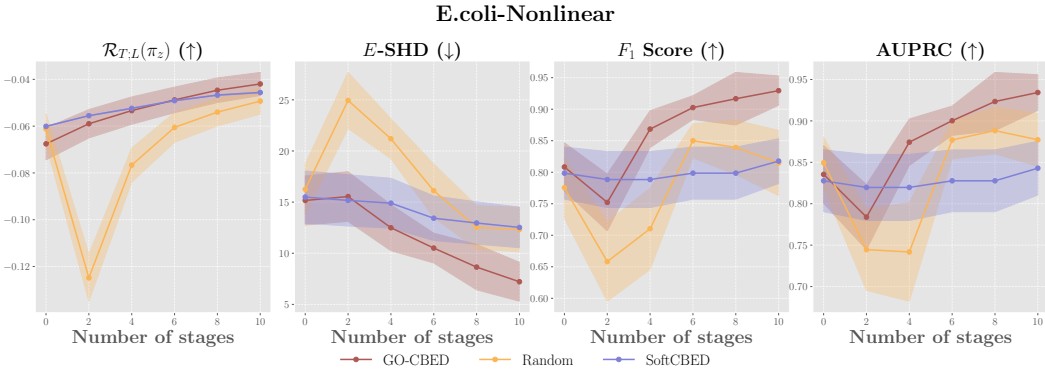

Figure 19: Causal discovery performance on nonlinear *E. coli* gene regulatory networks. GO-CBED performs better or comparatively in terms of uncertainty reduction ($\mathcal{R}_{T;L}$), structural recovery ($\mathbb{E}$-SHD), and structural accuracy ($F_1$-score and AUPRC) compared to all baselines. Shaded regions indicate $\pm 1$ standard error across 10 random seeds.

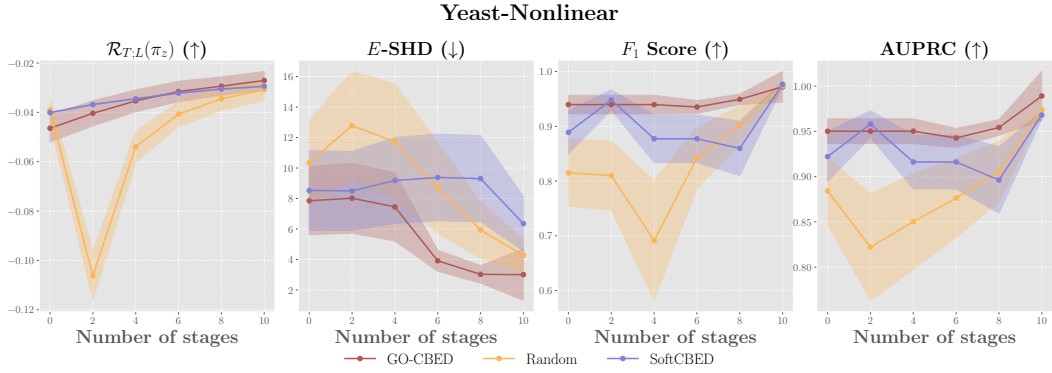

Figure 20: Causal discovery performance on nonlinear Yeast gene regulatory networks. GO-CBED performs better or comparatively in terms of uncertainty reduction ($\mathcal{R}_{T;L}$), structural recovery ($\mathbb{E}$-SHD), and structural accuracy ($F_1$-score and AUPRC) compared to all baselines. Shaded regions represent $\pm 1$ standard error across 10 random seeds.

metrics. Interestingly, even the random policy—when paired with its associated trained variational posteriors—achieves competitive performance, highlighting the advantage variational posteriors bring to causal learning.

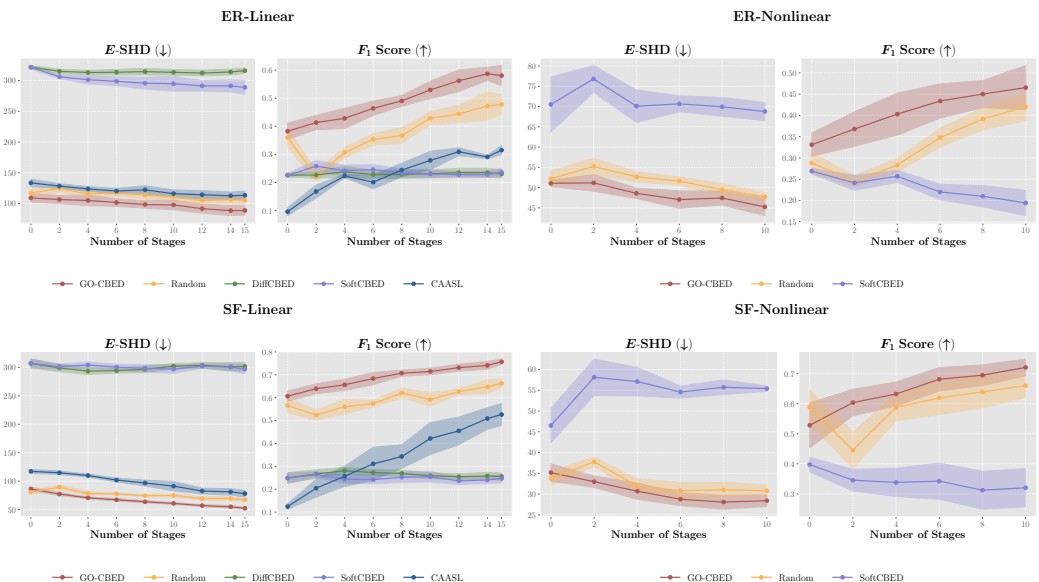

Figure 21: Performance comparison on synthetic SCMs, with each method using its originally proposed posterior inference approach. GO-CBED consistently outperforms all baselines across both linear and nonlinear settings, demonstrating the advantage of jointly optimizing policy and posterior networks. Shaded regions represent $\pm 1$ standard error across 10 random seeds.

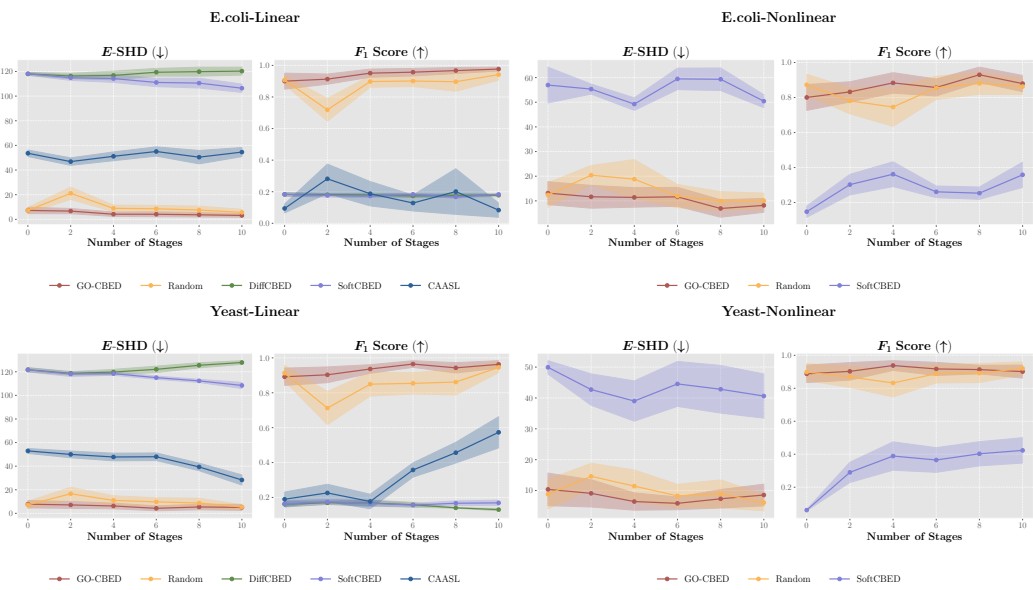

Figure 22: Performance comparison on semi-synthetic gene regulatory (*E. coli* and Yeast) networks, with each method using its originally proposed posterior inference approach. GO-CBED demonstrates strong performance on causal tasks in biologically inspired settings. Shaded regions represent $\pm 1$ standard error across 10 random seeds.

### D.5 DISTRIBUTIONAL SHIFT IN OBSERVATION NOISE

We evaluate the robustness of GO-CBED's policy and posterior networks under distributional shifts in observation noise. At deployment, the noise variance $\sigma_i^2$ is sampled from an inverse Gamma distribution, $\sigma_i^2 \sim \text{InverseGamma}(10, 1)$, in contrast to the fixed variance ($\sigma_i^2 = 0.1$) assumed during training. For comparison, we include a random intervention policy baseline, paired with a posterior network trained specifically on data with the shifted noise distribution.

We first focus on causal reasoning tasks, with ER and SF graph priors over 10-node networks. As shown in Figure 23, GO-CBED consistently outperforms the random baseline, demonstrating the robustness of both its policy and posterior networks in the presence of heteroskedastic noise. In the causal discovery setting (Figure 24), GO-CBED maintains strong performance, demonstrating its reliability across multiple causal tasks and noise conditions.

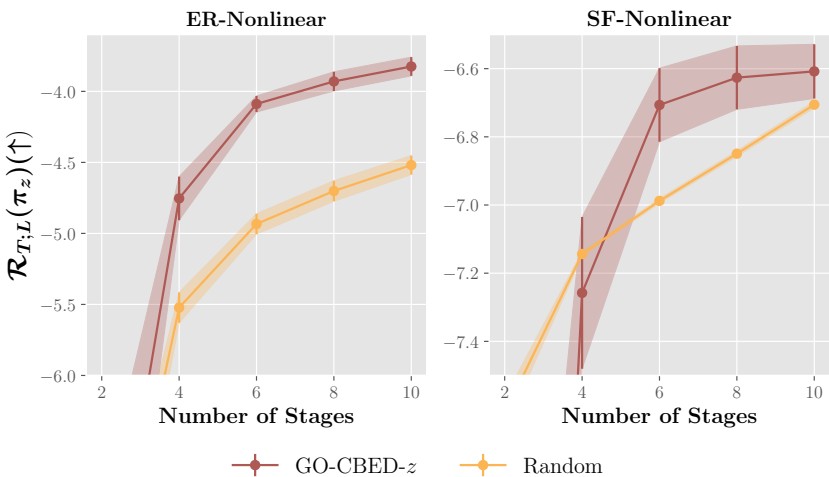

Figure 23: Evaluation of GO-CBED under a distributional shift in observation noise at deployment for causal reasoning tasks. GO-CBED consistently outperforms the random baseline that uses posterior networks trained under the shifted noise, demonstrating the robustness of its jointly optimized policy and posterior networks.

### D.6 PRIOR MIS-SPECIFICATION

To examine the effect of prior misspecification, we also conduct experiments on a $d = 10$ system where the prior assumes an Erdős–Rényi (ER) graph with expected number of edges equal to 2, while the ground-truth graphs are generated from an ER model with expected number of edges equal to 1. As shown in Fig. 25, the GO-CBED-$z$ policy continues to outperform the baselines under this mismatch.

### D.7 CAUSAL REASONING WITH SERGIO SIMULATOR

To evaluate GO-CBED under biologically realistic conditions, we conducted experiments using the SERGIO simulator (Dibaeinia & Sinha, 2020), which generates single-cell gene expression data through stochastic differential equations modeling transcriptional regulation.

**Experimental Setup.** We simulate Scale-Free gene regulatory networks with $d = 10$ genes and expected degree 2. SERGIO parameters are set as follows: basal production rates $\sim \text{Uniform}(1.0, 3.0)$, interaction strengths $\sim \text{Uniform}(1.0, 5.0)$, Hill coefficient $= 2.0$, and decay rate $= 0.8$. The simulator incorporates both intrinsic biological stochasticity and technical noise from the 10x Chromium platform (outlier, library size, and dropout effects). We evaluate on two causal queries: (1) $z = \{X_1, X_2 \mid \text{do}(X_6 = 0)\}$ and (2) $z = \{X_4, X_9 \mid \text{do}(X_5 = 0)\}$. For each query, policies are trained on data simulated from the above prior and evaluated on 10 independently sampled

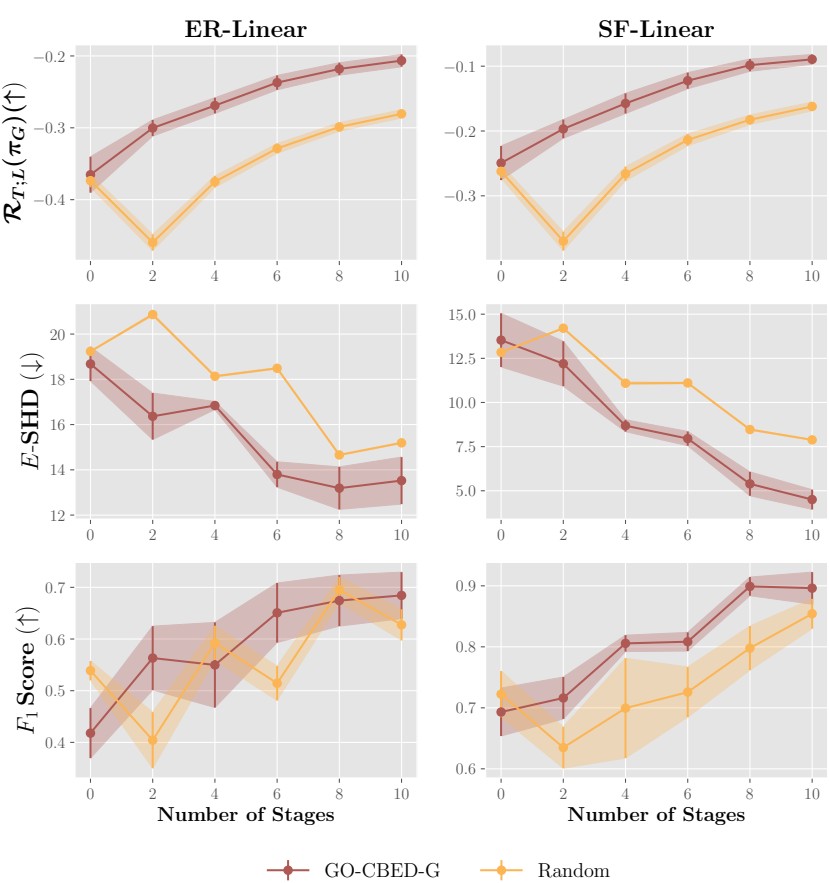

Figure 24: Evaluation of GO-CBED under a distributional shift in observation noise at deployment for causal discovery tasks. GO-CBED consistently outperforms the random baseline that is using posterior networks trained under the shifted noise, demonstrating the robustness of its jointly optimized policy and posterior networks.

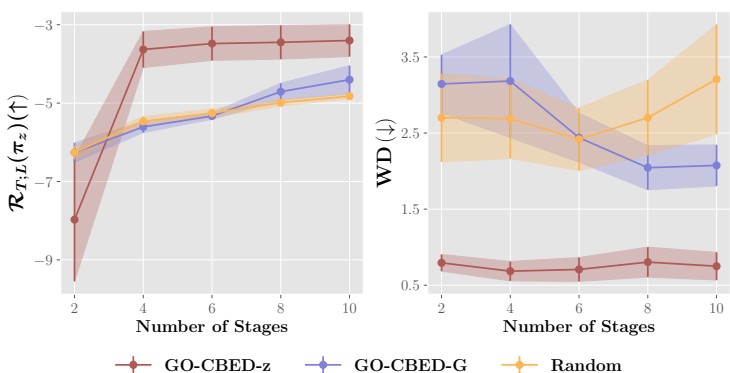

Figure 25: Effect of prior misspecification in the ER-graph setting.

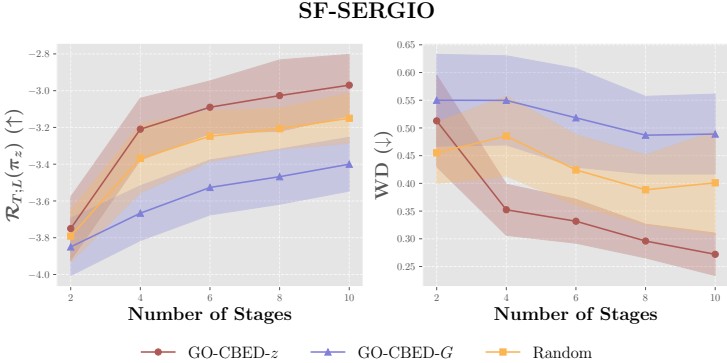

Figure 26: Performance comparison on Scale-Free graphs ($d = 10$, $T = 10$) with SERGIO simulated mechanisms on causal reasoning tasks. GO-CBED-$z$ outperforms baselines on both prior-omitted EIG lower bound (left, higher is better) and Wasserstein distance (right, lower is better). Shaded regions: $\pm 1$ standard error over 10 environments.

ground-truth SCMs. Each evaluation consists of $T = 10$ sequential stages with 10 interventional samples per stage.

**Results.** Figure 26 shows that GO-CBED-$z$ outperforms both GO-CBED-$G$ and random baselines. While variance is higher due to SERGIO's realistic stochastic dynamics, the consistent advantage across both metrics demonstrates that goal-oriented policy remains effective under complex biological noise.

# E    LLM USAGE

We used LLMs only for editing grammar, wording, and clarity of the written text. They were not used for ideation, methods, analysis, or drafting. All scientific content is by the authors, who take full responsibility.

