# OpenReview forum: "Goal-Oriented Sequential Bayesian Experimental Design for Causal Learning"
_ICLR.cc/2026/Conference — Submitted to ICLR 2026_

### Official Review · Reviewer_6rGY · 2025-10-31

**Soundness:** 3
**Presentation:** 3
**Contribution:** 3
**Rating:** 6
**Confidence:** 3

**Summary:**

The paper proposes GO-CBED, a goal-oriented, non-myopic Bayesian framework for sequential causal experimental design. Instead of maximizing information about the entire causal model, GO-CBED directly maximizes expected information gain (EIG) on user-specified causal quantities of interest (QoIs) (e.g., a particular interventional effect). The authors (i) cast sequential design as an RL policy that plans full intervention sequences; (ii) derive a variational lower bound on the EIG and learn it jointly with the policy; and (iii) instantiate an amortized transformer policy (permutation-aware) together with normalizing-flow posteriors for flexible QoIs. Empirically, GO-CBED outperforms baselines on synthetic SCMs and semi-synthetic gene-regulatory networks, with the largest gains when budgets are tight and mechanisms are nonlinear.

**Strengths:**

Originality. The paper presents a clear shift from model-centric to goal-centric BOED in causal settings. Whereas prior work either targeted full-model learning or was myopic, this paper unifies goal-orientation with non-myopic planning via RL. The variational EIG bound specialized to causal QoIs and combined with amortized policies is a fresh, well-motivated combination.

Theory & method. The EIG variational lower bound is clean and justified with the KL gap derivation, giving a tractable training objective while remaining faithful to the goal quantity. The use of normalizing flows to capture potentially multi-modal posteriors over effects is methodologically apt for structural uncertainty. The policy architecture respects permutation symmetries and handles discrete targets/continuous values (Gumbel-softmax head), which is thoughtful design for SCM variables.

Empirical significance. The manuscript delivers strong and consistent gains on causal reasoning under nonlinear mechanisms and on semi-synthetic GRNs, which are settings that mirror real experimental constraints. Notably, policies trained for structure learning underperform when the evaluation is a targeted effect, underscoring the value of aligning design with downstream goals.

Presentation clarity. Problem setup (SCMs, interventions, goal-oriented posterior over QoIs), objective (Eq. (4)), and training loop (Algorithm 1) are presented in a reader-friendly flow. Figures juxtapose “full-model” vs “goal-oriented” choices effectively (Fig. 1, Fig. 2–6).

**Weaknesses:**

Limited real-data validation. The results are synthetic or semi-synthetic (DREAM). While this is appropriate for a first step, I would recommend having a small real experimental case study (e.g., a wet-lab perturbation subset or an A/B platform with interventional logs) to materially strengthen the paper’s practical impact. Even a retrospective off-policy evaluation using logged interventions would be valuable.

Lack of ablation studies. The gains of the proposed method may stem from different sources (a) non-myopia, (b) goal-orientation, and (c) posterior flexibility. It would be helpful to include factorized ablations: (a) myopic vs non-myopic (same QoI objective), (b) goal-oriented vs model-oriented (same planner), and (c) flows vs simpler posteriors (e.g., Gaussian/mixture). To provide actionable insights on how the proposed framework could be effectively implemented, I recommend the authors additionally report sensitivity to the policy network depth, attention alternation, and Gumbel-softmax temperature.

Bound tightness. Theorem 4.1 ensures a lower bound, but the gap between $R_T$ and $R_{T;L}$ is not quantified. The authors may consider simple controlled settings (small DAGs with analytic posteriors) to measure bound tightness, beyond the linear fixed-graph toy in Section 5.1, and report how training correlates with true EIG.

Computational scalability. The method simulates many environments per step and trains both policy and posterior nets. It is useful to include wall-clock, GPU hours, and memory vs $d$, $T$, and per-stage intervention budget. Also, provide a throughput metric for the promised real-time deployment (ms per decision). Without such analyses, “constant complexity at deployment” risks sounding primarily qualitative.

Assumptions & robustness. Experiments primarily assume causal sufficiency and correct model class. Please probe robustness to misspecified priors, latent confounding, mechanism shift (beyond observation-noise shift), and heterogeneous intervention costs, all of which are common in practice. Even if some are left to future work, a compact stress test would be informative.

General QoIs & multi-target interventions. Most experiments focus on interventional effects and ($z=G$). Since the framework is billed as general, I suggest add at least one counterfactual or policy value QoI example, and a small demonstration of multi-target or soft interventions

**Questions:**

1.	Can you provide quantitative bound-gap measurements (true EIG vs variational bound) on small DAGs where posteriors are tractable, across training? Does tighter $q_{\lambda}$ (e.g., more flow layers) reliably improve policy quality?
2.	Could you add a controlled comparison where the only change is replacing the non-myopic policy with a greedy/myopic planner but keeping the same QoI objective and posterior family, to isolate the planning benefit?
3.	How sensitive is GO-CBED to prior over graphs and mechanisms? Any evidence or theory on graceful degradation if the prior is biased (e.g., wrong sparsity or mechanism class)?
4.	Many labs face heterogeneous costs per target and per perturbation magnitude. How would you incorporate cost-penalized EIG or budget constraints into the objective and policy?
5.	What are the largest $d$ and $T$ you trained successfully? Please report compute (GPU hours), RAM, and training curves vs problem size; also share inference latency (forward-pass ms) to substantiate the real-time claim.
6.	Any preliminary results for counterfactual QoIs or policy value estimation QoIs? If not, what modifications to $q_\lambda(z|\cdot)$ are needed?
7.	Why RealNVP over, say, MAF/Glow or neural spline flows? Did you observe training instabilities or mode-dropping with certain flow depths/widths?
8.	Since many labs adjust their budgets mid-campaign, how brittle is a policy trained for $T$ if deployed at $T'\neq T$? Any empirical robustness or simple re-planning heuristic?

---

> ### Author Response · Authors · 2025-11-21
> **Rebuttal by Authors (Part 1/3)**
>
> We sincerely thank the reviewer for the thorough and constructive review, and we are encouraged by the positive assessment of our work's originality, methodological soundness, and empirical significance. We detail our response below.
>
> ### W1: Limited real-data validation.
> We appreciate this suggestion. In response, we have added new experiments using the SERGIO simulator [1], which generates biologically realistic single-cell gene expression data. GO-CBED maintains strong performance advantages in these more complex settings. For detailed results, please see our response to Reviewer 1Qqi's W1 & Q2.
>
> [1] Dibaeinia, P., & Sinha, S. (2020). SERGIO: a single-cell expression simulator guided by gene regulatory networks. Cell systems, 11(3), 252-271.
>
> ### W2&Q2: Ablation studies (myopic vs. non-myopic)
>
> In principle, we could replace our non-myopic policy with a greedy planner to isolate the benefit of forward planning. However, a fair comparison is challenging because a myopic approach requires full re-inference at every stage after observing new data. While the graph posterior can be updated efficiently via our trained variational distribution, re-inferring the mechanism parameters $\theta$ at each stage is computationally expensive (requiring MCMC or variational inference) and scales linearly with horizon $T$. More critically, the accumulated approximation errors and optimization noise from stage-wise re-inference would confound the comparison, making it difficult to attribute performance differences purely to myopic vs. non-myopic planning.
>
> ### W3&Q1: Bound-gap measurements and tightness analysis
>
> Thank you for the question. We refer to the experiment in Appendix D.1 as a relevant illustration, where we compare our variational estimator to the nested Monte Carlo (NMC) baseline in a small 4-node linear Gaussian DAG with $T=1$. In this setting, we isolate the question of bound tightness by evaluating the expected information gain (EIG) directly without policy optimization.
>
> As shown in the example, GO-CBED achieves competitive performance using only $10^4$ samples, though the remaining gap can be difficult to close due to limitations in the flow architecture and the number of layers. In contrast, NMC requires $10^6$ to $10^7$ inner samples (and $5,000$ outer samples) to achieve low bias near the boundary of the intervention space. This demonstrates that the variational lower bound can be quite tight on small DAGs, even at modest training budgets, whereas traditional NMC becomes increasingly impractical in higher-node settings due to its steep computational demands.
>
> Regarding expressiveness, in principle, increasing the number of flow layers can tighten the bounds by improving posterior flexibility. However, deeper flow networks introduce more parameters, which can hurt stability and sample efficiency under fixed computational budgets. Empirically, we find that 3–4 flow layers provide a good tradeoff between expressiveness and training efficiency.
>
> Overall, our variational estimator offers substantial sample savings relative to NMC even when the posterior is analytically tractable, and the resulting bound is tight enough to reliably identify optimal designs.
>
> ### W4&Q5: Computational scalability
>
> We provide the requested computational metrics below:
>
> **Largest scale tested**: d=50 nodes, T=20 stages for causal discovery. Table 3 shows that GO-CBED maintains strong performance even at this scale, achieving the best AUPRC across both graph types.
>
> __Table 3: Causal Discovery on 50-Node Scale-Free Graphs with Linear Mechanisms (T=20)__
>
> | CBED Methods | $\mathbb{E}$-SHD ($\downarrow$)  | $F_1$-score ($\uparrow$)       | AUROC ($\uparrow$)             | AUPRC ($\uparrow$)             |
> | ------------ | ----------------- | ----------------- | ----------------- | ----------------- |
> | CAASL        | $227.60 \pm 4.25$ | $0.291 \pm 0.035$ | $0.933 \pm 0.005$ | $0.596 \pm 0.019$ |
> | Random       | $201.09 \pm 4.65$ | $0.668 \pm 0.022$ | $0.795 \pm 0.016$ | $0.693 \pm 0.017$ |
> | GO-CBED      | $179.06 \pm 3.93$ | $0.706 \pm 0.01$  | $0.822 \pm 0.01$  | $0.726 \pm 0.014$ |
>
> **Training time**: Several to tens of hours on NVIDIA A40 GPUs depending on problem size (d=10-50, T=10-20). Memory usage remains under 16GB across all settings. We note our implementation is not optimized for efficiency; training time could be substantially reduced with engineering effort (e.g., parallelization, mixed precision).
>
> **Deployment latency** (forward pass):
> | Problem size      | d=10, T=10 | d=20, T=10 | d=30, T=15 | d=50, T=20 |
> | ----------------- | ---------- | ---------- | ---------- | ---------- |
> | Latency (ms)| 8.97       | 8.78       | 8.96       | 17.87      |

---

> ### Author Response · Authors · 2025-11-21
> **Rebuttal by Authors (Part 2/3)**
>
> ### W5&Q3: Sensitivity to prior misspecification
>
> Please see our detailed response to Reviewer fTBu's Q1 for an empirical example of prior misspecification. In general, the sensitivity of GO-CBED reflects the usual tradeoff in Bayesian design: the prior should be broad enough to place the ground truth well within its support, but an excessively wide prior expands the trajectory space and can make the variational lower bound harder to optimize under a fixed training budget.
>
> That said, because we perform a sequence of experiments, the posterior is updated with a trajectory of data, allowing the belief to contract toward the ground truth even when the initial prior is somewhat biased. Our empirical results (including the misspecified ER example) show that GO-CBED exhibits graceful degradation under moderate mismatch and continues to outperform baselines, suggesting that the method is reasonably robust to prior misspecification in both graph sparsity and mechanism parameters.
>
> ### W5&Q6: General QoIs & multi-target interventions
>
> Our framework handles arbitrary causal queries through the flexible function $\boldsymbol{z} = H(\boldsymbol{\mathcal{M}}; \epsilon_{\boldsymbol{z}})$ and requires no architectural modifications for counterfactual or policy value QoIs—only the specification of $H$ and posterior family $q_{\lambda}(z|·)$ changes.
> In settings with ground-truth SCM access (our experiments), counterfactual quantities can be expressed using interventional and observational distributions. For example, counterfactual outcomes $p(X_i | do(X_j), X_k=x_k)$ can use the same normalizing flow posteriors as interventional effects, as both represent continuous outcome distributions.
> We focused on interventional effects as the most common causal query in experimental design. Explicit counterfactual experiments would strengthen our generality claims and are planned for future work.
>
> ### Q4: Incorporating heterogeneous costs
>
> Thank you for the insightful question. We agree that accounting for heterogeneous costs is critical in many real-world settings. One natural extension of our framework is to optimize a cost-penalized acquisition objective of the form:
> $\mathrm{EIG}(\pi) - \lambda \cdot \mathbb{E}_{\pi}[\mathrm{Cost}]$,
> where λ is an exchange rate that quantifies the value of one bit of information relative to cost. This requires the user to define a meaningful tradeoff between informativeness and resource expenditure (i.e., what one bit of information is “worth” in their experimental context).
>
> Alternatively, the problem can be formulated as a multi-objective optimization, jointly maximizing EIG and minimizing cost. In this setting, one could learn a family of Pareto-optimal policies that trace out the frontier between information gain and cost. This allows the user to select a policy best aligned with their budget or experimental constraints post hoc.
> Our framework is compatible with both formulations, and we believe this is a promising direction for future work.
>
> ### Q7: Why RealNVP over MAF/Glow or neural spline flows?
>
> We agree that MAF, Glow, and neural spline flows are valid alternatives. In our setting, the key consideration is that the QoI dimension is very small, which is one of the motivations for focusing on QoI rather than PoI, as amortized posterior inference becomes much simpler and more stable in low dimensions. RealNVP provides sufficient expressivity in this regime. In addition, we selected RealNVP mainly for computational simplicity: it has a simpler Jacobian structure and fewer sequential dependencies than autoregressive flows, which leads to faster training and lower memory overhead -- important for the inner-loop variational inference that must be repeated many times during policy learning. While more complex flow architectures could also be used or even achieve a tighter lower bound, RealNVP was already effective and efficient for our needs

---

> ### Author Response · Authors · 2025-11-21
> **Rebuttal by Authors (Part 3/3)**
>
> ### Q8: Robustness to changing experimental budgets
>
> In our experiments, we evaluated robustness to shortened horizons by training policies for the full horizon $T$ and testing them at intermediate stages $t < T$. We found that GO-CBED generally begins to outperform baselines from the middle stages onward, with performance steadily improving as $t$ approaches $T$. These results suggest that the learned policy remains effective under early stopping, demonstrating robustness to mid-campaign budget changes.
>
> For scenarios with highly uncertain budgets, a natural extension is to emphasize earlier stages during training. Our current architecture uses max pooling over stage-wise embeddings to form a single representation for posterior learning, which does not naturally allow stage-wise reweighting. However, this can be addressed with a minor architectural modification: replacing max pooling with a learnable weighted aggregation (e.g., attention mechanism) over stage embeddings. This would preserve the overall encoder-posterior structure while allowing the model to adaptively prioritize earlier stages when beneficial.

---

> > ### Author Response · Authors · 2025-11-28
> > **Follow-up on Rebuttal**
> >
> > Dear Reviewer `6rGY`,
> >
> > We sincerely appreciate your detailed feedback, which greatly helps us improve our work.
> >
> > In response to your concerns, we have conducted SERGIO simulator experiments for realistic validation (`Appendix D.7`), provided detailed computational scalability and latency benchmarks for larger graphs up to $d=50$, and evaluated robustness under prior misspecification (`Appendix D.6`).
> >
> > We believe these enhancements address the key points you raised and provide stronger support for the paper’s contributions. If further clarification would be helpful, please let us know.
> >
> > Thank you again for your time and valuable suggestions!
> >
> > Best regards,
> >
> > The Authors of Submission 14524

---

### Official Review · Reviewer_1Qqi · 2025-10-31

**Soundness:** 3
**Presentation:** 3
**Contribution:** 2
**Rating:** 4
**Confidence:** 2

**Summary:**

The paper proposes a new algorithm for goal conditioning for causal experimental design. This has potentially important applications in scientific domains such as biological research.
More concretely, the paper builds on CBED proposed in [1] and the proposed GO-CBED is a goal conditioned version of that algorithm.

Overall the paper is easy to read and has a lot of additional and interesting experiments in the appendix for which there was no space in the main text.

[1] Tigas, Panagiotis, et al. "Interventions, where and how? experimental design for causal models at scale." Advances in neural information processing systems 35 (2022): 24130-24143.

**Strengths:**

It is addressing and important problem and generally it makes sense to use targeted approaches in practice. It is just not clear if that is then evaluated fairly.

**Weaknesses:**

Where are results on the Sergio environment [1] similarly to [2]. The dream datasets has significant issues and is mostly used because it is very well established and thus has a form of legacy status where new papers need to compare too but in order to have some meaningful statements you need to provide results with a simulator e.g. Sergio.

General statements sometimes seem not to be correct e.g. "GO-CBED outperforms existing methods" It would be better to have more nuanced statements. i.e. at least say in which tasks or in which settings since  in all its generality this statement is probably wrong.

"directly targeting causal queries is particularly advantageous compared to targeting the full causal graph. " This is an apple to pear comparison. It is likewise known in other disciplines that if you do not care about the causal structure then you are better of only learning the causal features e.g [3] but there are many more papers and this should not be surprising that the full graph fulfils a different purpose and should not be compared to approaches which are specialized and then compared a universal approach on the specialization task.



[1] Payam Dibaeinia and Saurabh Sinha. Sergio: a single-cell expression simulator guided by gene
regulatory networks. Cell systems, 2020.
[2] Annadani, Yashas, et al. "Amortized active causal induction with deep reinforcement learning." Advances in Neural Information Processing Systems 37 (2024).
[3] Kim, Jang-Hyun, et al. "Large-Scale Targeted Cause Discovery via Learning from Simulated Data." Transactions on Machine Learning Research.

**Questions:**

Can you state what the specific causal queries were you used for training in each experiment? I actually could not find this specification for any of the experiments but I might have overlooked it. Where is it clearly stated?

Can you compare and provide experiments with the Sergio Simulator?

---

> ### Author Response · Authors · 2025-11-21
> **Rebuttal by Authors (Part 1/2)**
>
> Thank you for your thoughtful review and for recognizing the importance of our problem setting! In direct response to your valuable suggestions, we have conducted new experiments using the SERGIO simulator and revised our claims throughout the paper to be more specific about the settings where our method excels. Below, we address each of your concerns in detail.
>
> ### W1&Q2: Evaluation on the SERGIO Simulator
>
> Thank you for your suggestion. In direct response, we have conducted new experiments using SERGIO (Single-cell Expression of Genes In Silico) on the causal reasoning tasks.
>
> __Experimental Setup__:
>
> We use scale-free graphs (10 nodes, expected degree = 2) with biologically realistic SERGIO parameters: basal rates $\sim \mathrm{Uniform}(1.0, 3.0)$ , interaction strengths $\sim \mathrm{Uniform}(1.0, 5.0)$, Hill coefficient $= 2.0$, and decay rate $= 0.8$. The simulator incorporates both intrinsic biological stochasticity and technical noise from the 10x Chromium platform (outlier, library size, and dropout effects).
>
> We evaluate on two causal queries: (1) predicting the expression of genes {1, 2} under intervention on gene 6, and (2) predicting the expression of genes {4, 9} under intervention on gene 5. For each query, we train policies using data simulated from the above prior, then evaluate on 10 independently sampled ground-truth SCMs (graph structure and parameters drawn from the same prior). Each evaluation consists of 10 sequential stages, with 10 interventional samples collected per stage following each policy's decisions.
>
> Results (Table 2) demonstrate that GO-CBED-$z$ outperforms both baselines on both metrics. It achieves the highest variational lower bound and the lowest Wasserstein distance to the true interventional distribution. While the variance is higher due to SERGIO's realistic stochastic dynamics, the consistent advantage across both metrics indicates that goal-oriented design remains effective even with complex biological noise.
>
>
> __Table 2__: Causal Reasoning on SERGIO Simulator ($T=10$)
> | Method               | $\boldsymbol{\mathcal{R}_{T;L}(\pi)} ( \uparrow )$   | $\mathbf{WD} ( \downarrow )$            |
> | --------             | --------                                | --------                 |
> | GO-CBED-$z$          |  $-2.97 \pm 1.36$                   | $0.272 \pm 0.15$     |
> | GO-CBED-$G$          |  $-3.40 \pm 1.17$                   | $0.489 \pm 0.28$     |
> | Random               | $-3.15 \pm 1.09$                    | $0.451 \pm 0.27$     |
>
> ### W2: More nuanced claims and the value of our comparison
> We thank the reviewer for this feedback and for pointing us to [3]. We have revised the manuscript throughout to provide more precise statements about our method's performance. For example, we now specify that "GO-CBED achieves strong performance on query-focused experimental design tasks, particularly in settings where the downstream objective involves specific causal quantities of interest," rather than making broad claims about universal superiority.
>
> We appreciate the reviewer's point about the "apple-to-pear" comparison and that targeted approaches should outperform general ones when evaluated on specialized tasks, as demonstrated by [3] for observational causal discovery. We want to clarify that (1) our work addresses a different problem: sequential interventional experimental design. While [3] learns causal parents from fixed observational data, our framework must actively choose which interventions to perform and plan non-myopically over future stages. (2) GO-CBED is a flexible framework where users specify their quantity of interest, whether it's a specific causal effect, a subset of edges, or the entire graph. GO-CBED-$G$ and GO-CBED-$z$ are not "general vs. specialized" methods, but rather two instantiations of the same framework optimizing different Bayesian objectives.
>
> The comparison demonstrates the practical efficiency cost of the traditional "learn the full model first" approach when the actual goal is to answer a specific causal query. We fully agree that when the objective is complete graph recovery, structure-learning methods are the appropriate comparison and should be evaluated on structure-learning metrics (which we provide in Section 5.3 and Appendix D). We have revised the paper to make these distinctions clearer and properly cite [3] in the Detailed Related Work section.

---

> > ### Author Response · Authors · 2025-11-21
> > **Rebuttal by Authors (Part 2/2)**
> >
> > ### Q1: Specification of Causal Queries Used for Training
> >
> > Thank you for this question. The specific causal queries are described in Appendix C.3 (for general setup) and Appendix D.3 (for additional examples), but we agree that this information should be more explicit. Here we provide a clear summary:
> >
> > __Causal Reasoning Experiments (Section 5.2)__:
> > * Erdős–Rényi (ER): $\mathbf{z} = \{X_{10}, X_{18}\} \mid \text{do}(X_2 \sim \mathcal{N}(4, 2^2))$
> > * Scale-Free (SF): $\mathbf{z} = \{X_4, X_{10}\} \mid \text{do}(X_{11} \sim \mathcal{N}(4, 2^2))$
> > * E.coli Gene Regulatory Network: $\mathbf{z} = {X_6, X_{8}} \mid \text{do}(X_{7} \sim \mathcal{N}(4, 2^2))$
> >
> > Additional queries on Yeast networks and diverse structures are detailed in Appendix D.2. The policy is trained on the specified query type with graphs and parameters sampled from the prior distributions described in Appendix C.3.
> >
> > __SERGIO Experiments (added during rebuttal)__:
> >
> >
> > For the SERGIO experiments, the causal queries include
> > $\mathbf{z} = \{X_1, X_{2}\} \mid \text{do}(X_{6} = 0))$ and
> > $\mathbf{z} = \{X_4, X_{9}\} \mid \text{do}(X_{5} = 0))$.
> >
> > We have revised the manuscript to make this information more explicit in the main text and have added a summary table in Appendix C.4 for easy reference. The table includes both the specific causal queries and the corresponding ground truth graphs used in each experiment.

---

> > > ### Author Response · Authors · 2025-11-28
> > > **Follow-up on Rebuttal**
> > >
> > > Dear Reviewer `1Qqi`,
> > >
> > > Thank you for your thoughtful review and constructive suggestions.
> > >
> > > Following your suggestions, we have conducted additional experiments using the SERGIO simulator (`Appendix D.7`) to validate our method on realistic data, explicitly listed all causal queries in a new summary table (`Appendix C.4`), and revised our claims to be more nuanced regarding the comparison with structure learning methods.
> > >
> > > We hope these additions and clarifications have resolved your concerns. If you find our response satisfactory, we would be grateful for your reconsideration of the score. If any questions remain, we are happy to provide further clarification.
> > >
> > > Thank you again for your careful review and helpful feedback!
> > >
> > > Best regards,
> > >
> > > The Authors of Submission 14524

---

### Official Review · Reviewer_cF4q · 2025-10-31

**Soundness:** 2
**Presentation:** 3
**Contribution:** 3
**Rating:** 6
**Confidence:** 3

**Summary:**

The paper proposes a new method for Bayesian experimental design. The method is the first to integrate goal-oriented reasoning with a non-myopic decision-making framework..The approach relies on amortized learning. The authors propose to jointly learn the amortized variational posterior of the objective and the amortized policy. GO-CBED presents favorable performance when compared to baselines on a set of synthetic and semi-synthetic problems of varying sizes.

**Strengths:**

1. The paper tackles the important problem of efficient experimental design, optimizing the full sequence of interventions with respect to the query of interest rather than using a greedy strategy, and presents a novel solution.
2. The paper is well-executed, the descriptions are extensive, and the details provided allow reproducibility to a high extent.
3. The paper is mostly clear and well-written.
4. Experiments are clearly presented.

**Weaknesses:**

1. The experimental setup is unclear. I could not find information on what priors over graphs and mechanisms were used for network training.
2. It seems to me that the method is only evaluated in an in-distribution setting. Lack of discussion and experimental evaluation for out-of-distribution/real-world examples severely limits the potential impact of the work and its significance.
3. The method dropped the acyclicity constraint used by other causal methods, without discussion. This could affect the effectiveness of rejection sampling of DAGS from the posterior and reduce the practical utility of the approach for large graphs. How does the number of samples required to obtain a DAG change with the size of the graph?
4. The description of the method is unclear. How is the variational posterior of z trained in the causal reasoning setup? How is query z encoded? Is there a need to train a separate posterior model for each query and graph size?

**Questions:**

1. Is it possible to ablate the influence of non-myopticity from other components of your method? I.e., is it possible to train a greedy policy using the proposed approach with minimal changes?
2. Line 458 “biologically-inspired networks demonstrate GO-CBED’s ability to handle the complex dependencies and noise typically encountered in gene regulatory systems” — Does the evaluation use a biologically inspired mechanisms? This is unclear even after reading the appendix.
3. How many different models (pretraining runs)  were required to provide results from the experimental section? How much data is required for training, and how long does it take?
4. What is the difference between GO-CBED-\theta and GO-CBED-G? Is the structure in section 5.1 known?
5. Can a trained model generalize to different graph sizes?
6. The Authors mention that the method enables “real-time decision-making”. Is it really a practical benefit? Could the Authors provide an example of the experimental design problem that is time-sensitive?
7. Why has the lower bound objective been used to compare methods in Figure 2? Could the Authors provide a comparison between methods in terms of the  Wasserstein Distance?
8. How is experimentation horizon T affecting the effectiveness of the method? Which performs better, the model trained with horizon t0 or the model trained with a longer horizon stopped at intervention t0?
9. There seems to be an error in line 1095: "T=1500 when d=30"

---

> ### Author Response · Authors · 2025-11-21
> **Rebuttal by Authors  (Part 1/3)**
>
> Thank you for your insightful feedback! We are encouraged that you found our approach novel and well-executed, with clear presentation and extensive details. We address your concerns below.
>
> ### W1: Unclear experimental setup
>
> Thank you for this feedback. The complete experimental setup is detailed in Appendices C.2 and C.3. Key details:
>
> * Graph priors: Erdős-Rényi (edge probability scaled to O(d) expected edges), Scale-Free (Barabási-Albert preferential attachment), and DREAM gene regulatory networks.
> * Mechanism priors:
>   * Linear: $X_i = \boldsymbol{\theta}\_i\^\top \boldsymbol{X}\_{pa(i)} + b_i + \epsilon_i,$ with $\boldsymbol{\theta}_i \sim \mathcal{N}(0, 2)$, $b_i \sim \mathcal{U}(-1, 1)$, and $\epsilon_i \sim \mathcal{N}(0, 0.1^2)$.
>   * Nonlinear: 2-layer neural networks (8 hidden ReLU units) with standard normal weight priors.
>
> ### W2: Only in-distribution evaluation, lacking OOD/real-world experiments and discussion.
>
> We respectfully clarify that we have evaluated OOD robustness in multiple ways. In Appendix D.5, we tested policies trained with fixed observation noise under heteroskedastic noise at deployment, where GO-CBED maintained strong performance despite this shift.
>
> During rebuttal, we conducted additional OOD experiments. We evaluated prior misspecification by training on ER graphs expecting 2 edges/node and testing on graphs with 1 edge/node (__response to Reviewer fTBu Q1__), where GO-CBED continues to outperform baselines. We also added experiments using the SERGIO simulator with stochastic single-cell RNA-seq data and 10x Chromium noise characteristics (__response to Reviewer 1Qqi W1&Q2__). These results demonstrate robustness across noise distributions, prior misspecification, and biologically realistic data-generating processes.
>
> ### W3: The method lacks discussion of acyclicity constraints, which could affect DAG sampling efficiency for large graphs.
>
> Thank you for the question. We would like to clarify that for causal discovery tasks, we do incorporate acyclicity constraints following AVICI [1], treating them as a penalty term through Lagrange multiplier formulation during posterior training. We omitted these details in the main text because empirically, we found that adjusting the acyclicity penalty weight $\lambda$ had minimal impact on both the learned posterior quality and DAG sampling efficiency across our experimental settings. During inference, we use rejection sampling to ensure valid DAGs. In our experiments, the rejection rate remained manageable even for 50-node graphs, but we acknowledge that for very large graphs, more sophisticated approaches regarding the DAG constraints may be necessary, and we will discuss this as a direction for future work. Note that for causal reasoning tasks, acyclicity is not a concern as the posterior targets causal effects rather than graph structures.
>
> [1] Lorch, L., Sussex, S., Rothfuss, J., Krause, A., & Schölkopf, B. (2022). Amortized inference for causal structure learning. Advances in Neural Information Processing Systems, 35, 13104-13118.
>
> ### W4: Unclear variational posterior training, query encoding, and whether separate models are needed per query/size.
>
> Thank you for the question. The variational posterior approximates $p(\mathbf{z} \mid h_T)$, where $h_T$ is the observed trajectory and $z$ is the target causal quantity of interest. We would like to clarify that  **z is the target for density approximation, not an input**: it is simulated using causal model $\mathcal{M}$ that also governs $h_T$. This shared dependency enables learning the conditional density $q_{\lambda}(\mathbf{z} \mid f_{\phi}(h_T))$, where $f_{\phi}$ embeds the trajectory and a conditional normalizing flow with parameters $\lambda$ provides flexible posterior approximation.
>
> We currently train separate posteriors per query and graph size for computational efficiency. However, the framework naturally extends to: (1) joint queries: multiple $\mathbf{z}\_1, \mathbf{z}\_2, \ldots$ can be modeled jointly as $q\_{\lambda}(\mathbf{z}\_1, \mathbf{z}\_2, \ldots \mid f\_{\phi}(h\_T))$ if they share the same $\mathcal{M}$, and (2) size/structure generalization: encoding graph uncertainty in the $\mathcal{M}$ prior allows a single model to handle varying dimensions and topologies.

---

> > ### Author Response · Authors · 2025-11-21
> > **Rebuttal by Authors (Part 2/3)**
> >
> > ### Q1: Can you ablate non-myopic planning by training a greedy policy baseline?
> >
> > Thank you for this suggestion. While conceptually straightforward, training a greedy policy within our framework is computationally prohibitive and yields an unfair comparison.
> > A greedy policy requires full posterior inference (over graphs and parameters) after every single stage to select the next action, which is substantially more expensive than GO-CBED's amortized joint training across all stages. Moreover, greedy training compounds multiple error sources: variational approximation error, finite-sample estimation noise, and optimization instability at each step. Any performance degradation would be confounded by these implementation issues rather than reflecting the true value of non-myopic planning.
> >
> > ### Q2: Do evaluations use biologically realistic mechanisms? Line 458 is unclear.
> >
> > Thank you for highlighting this ambiguity. We clarify that DREAM experiments use real E. coli/Yeast networks but generic mechanisms (linear or neural network-based). To address this, we conducted new SERGIO experiments (response to Reviewer 1Qqi) with biologically realistic mechanisms using the SERGIO simulator.
> >
> > We have revised line 458 (now line 463) to "semi-synthetic networks with realistic biological topologies".
> >
> > ### Q3: How many models were trained? What are the data requirements and training times?
> >
> > We train one model per prior specification (graph type + mechanism type). Each run samples 5 environments per step and converges within 5,000-10,000 steps. For d ≤ 50 nodes, training takes several to tens of hours on NVIDIA A40 GPUs, depending on simulator complexity.  While this requires upfront computational investment, the amortized policy enables constant-time deployment for sequential decision-making, providing substantial efficiency gains over non-amortized methods that require expensive online optimization at each experimental stage.
> > Our current implementation trains separate models per prior for experimental clarity. However, a unified model trained across diverse priors could reduce training overhead. We leave this promising direction for future work.
> >
> > ### Q4: Difference between GO-CBED-θ and GO-CBED-G? Is the Section 5.1 structure known?
> >
> > GO-CBED-$\theta$ targets parameter uncertainty (reducing uncertainty over $\theta$), while GO-CBED-$G$ targets graph structure uncertainty (reducing uncertainty over $G$).
> > In Section 5.1, the graph structure is fixed and known—only $\theta$ is uncertain. This simplified setting enables analytical posterior computation and provides a clean comparison of goal-oriented (GO-CBED-$z$) versus parameter-oriented (GO-CBED-$\theta$) policies.
> >
> > ### Q5: Can a trained model generalize to different graph sizes?
> >
> > Yes, the transformer architecture can generalize across graph sizes. Our current experiments use fixed sizes for controlled baseline comparison, but the framework naturally supports size generalization.
> > By training with a prior over graph sizes (e.g., $d \sim \mathrm{Uniform}(10, 20, 30, 50)$), the policy learns size-invariant intervention strategies. The permutation-equivariant design enables this flexibility without architectural changes.
> >
> > ### Q6: Is "real-time decision-making" a practical benefit? Any time-sensitive examples?
> >
> > The key practical benefit of our approach is that, once trained, the design policy replaces expensive online Bayesian optimization with a single forward pass, enabling fast deployment. This is particularly valuable in sequential experiments where system dynamics evolve on comparable timescales to measurement intervals.
> > For instance, in pharmacokinetic (PK) studies [2], drug concentrations are measured over short time windows (e.g., hours), and experimenters must choose the next sampling time or dose within seconds to minutes. In genetic perturbation experiments (e.g., CRISPR knockouts), while individual experiments take longer, fast policy evaluation eliminates computational bottlenecks in automated platforms where throughput depends on rapid decision-making between perturbation rounds.
> >
> > [2] Ivanova, D. R., Foster, A., Kleinegesse, S., Gutmann, M. U., & Rainforth, T. (2021). Implicit deep adaptive design: Policy-based experimental design without likelihoods. Advances in Neural Information Processing Systems, 34, 25785–25798.

---

> > > ### Author Response · Authors · 2025-11-21
> > > **Rebuttal by Authors (Part 3/3)**
> > >
> > > ### Q7: Why is Figure 2 based on the lower bound? Can you add Wasserstein Distance comparison?
> > >
> > > Thank you for pointing this out. We clarify that Figure 2 uses the true EIG ($\mathcal{R}\_{T}$), not the variational lower bound ($\mathcal{R}\_{T;L}$). Because the graph is fixed and the mechanisms are linear Gaussian in this experiment, posteriors can be computed analytically via conjugacy, allowing direct Monte Carlo estimation of the exact EIG objective. We have corrected the y-axis label of Figure 2 to avoid confusion.
> > >
> > > We have also added Wasserstein distance (WD) evaluation as requested. The table below ($T=10$) shows WD between posterior predictive distributions from each policy and the ground truth. A complete evaluation is included in Appendix C.2. Policies achieving higher EIG consistently yield posteriors closer to ground truth, confirming the correlation between our optimization objective and downstream quality:
> > >
> > > | Method               |  $\mathbf{WD} ( \downarrow )$           |
> > > | -------------------- | ---------------------------------------- |
> > > | GO-CBED-$z$          |  $0.093 \pm 0.026$                   |
> > > | GO-CBED-$\theta$     |  $0.210 \pm 0.109$                 |
> > > | NMC                  |  $0.418 \pm 0.042$                   |
> > > | Random               |  $0.547 \pm 0.103$                   |
> > >
> > >
> > > ### Q8: How does horizon $T$ affect performance? Does training for $t_0$ outperform training for $T>t_0$ when evaluated at $t_0$?
> > >
> > > Thank you for this important question. Given fixed training samples, increasing horizon T makes trajectories sparser, which can slow the convergence of the variational lower bound.
> > > A model trained only to horizon $t_0$ typically outperforms a longer-horizon ($T>t_0$) model when both are evaluated at stage $t_0$. This is expected: the greedy policy optimizes specifically for that stage, while the non-myopic policy optimizes for the full trajectory. However, greedy policies are known to be suboptimal overall [3].
> > > In our experiments, we train policies for the horizon $T$ and evaluate at all intermediate stages $t<T$. While non-myopic policies may initially lag, they achieve higher EIG as $t$ approaches $T$, reflecting their global optimality. This validates the trade-off: short-term performance at early stages versus long-term cumulative gain.
> > >
> > > [3] Shen, W., & Huan, X. (2023). Bayesian sequential optimal experimental design for nonlinear models using policy gradient reinforcement learning. Computer Methods in Applied Mechanics and Engineering, 416, 116304. https://doi.org/10.1016/j.cma.2023.116304
> > >
> > > ### Q9: There seems to be an error in line 1095: "T=1500 when d=30"
> > > Thank you for catching this typo. The correct value is $T=15$, and we have corrected it.

---

> > > > ### Author Response · Authors · 2025-11-28
> > > > **Follow-up on Rebuttal**
> > > >
> > > > Dear Reviewer `cF4q`,
> > > >
> > > > We sincerely appreciate your positive assessment and detailed feedback on our work.
> > > >
> > > > Following your feedback, we have conducted additional experiments on prior misspecification (`Appendix D.6`), added Wasserstein distance metrics to validate our objective (`Appendix C.2`), and clarified the experimental setup (`Appendix C`).
> > > >
> > > > We hope these additions and clarifications address your concerns and help justify a stronger recommendation. If any questions remain, we are happy to provide further clarification.
> > > >
> > > > Thank you again for your thoughtful review and helpful suggestions!
> > > >
> > > > Best regards,
> > > >
> > > > The Authors of Submission 14524

---

### Official Review · Reviewer_fTBu · 2025-10-31

**Soundness:** 3
**Presentation:** 3
**Contribution:** 2
**Rating:** 4
**Confidence:** 4

**Summary:**

The paper proposes an algorithm GO-CBED for sequential experimental design. Experiments are selected to be informative in terms of certain causal targets, quantities of interest (QoIs), as opposed to, for example, estimating a full causal model. This setting has been studied before, but previously been addressed using greedy selection algorithms. The paper presents an approach, GO-CBED, which amortizes a non-myopic experimental design policy. A variational lower bound on the expected information gain is derived, which is utilized as training objective. Experiments on synthetic simulations (in one case a semi-synthetic example inspired by gene regulatory networks) provide evidence of the algorithms’ performance.

**Strengths:**

- The considered experimental design setting is practically relevant and (while not new) not yet extensively studied.
- Amortization of a policy can provide computational speedups at deployment time.

**Weaknesses:**

- All the major components (targeting experimental design for causal objectives setting; computing variational bounds on mutual information; policy amortization for active learning / experimental design) have already been quite well explored; the main contribution of the paper is to put them together.
- Experiments are mostly on synthetic settings (the only exception is the semi-synthetic GRN example).

**Questions:**

- I assume the synthetic graphs / mechanisms are sampled from a different distribution than what is used as a prior in the Bayesian model. Can you please confirm? If not, what happens if the prior is misspecified?

---

> ### Author Response · Authors · 2025-11-21
> **Rebuttal by Authors**
>
> Thank you for your constructive comments! We appreciate that the considered experimental setting is practically relevant and not yet extensively studied. We address your concerns below in detail
>
> ### W1: All the major components have already been quite well explored; the main contribution of the paper is to put them together.
>
> We thank the reviewer for the comment and would like to clarify that GO-CBED is not a simple aggregation of known components. While goal-oriented design, variational MI bounds, and amortized policies have each been explored individually, our contribution is the first framework that unifies goal-orientation with non-myopic sequential planning for arbitrary causal quantities of interest (QoIs). This requires addressing unique technical challenges:
>
> * Causal-Specific Design: Our framework handles the structural challenges unique to SCMs: uncertainty over discrete graphs ($G$) and graph-dependent continuous mechanisms ($\theta$), through permutation-aware policy architectures and specialized posterior families (edge-wise Bernoulli, normalizing flows).
> * Variational EIG Bound for Causal Quantities: We derive a new bound defined over causal functionals (e.g., interventional effects, causal structure learning) rather than model parameters. Existing MI bounds do not directly apply to these derived quantities.
> * Joint Training Architecture: We jointly train the variational posterior and long-horizon planner, which has not been done in prior causal BOED work. This co-adaptation is critical for learning policies that generate experiments informative for the posterior's current modeling capacity
>
> Empirically, this unified formulation enables capabilities that none of the prior components alone provide: methods targeting full-model learning or greedy goal-oriented baselines perform substantially worse on targeted causal queries, especially under long horizons or nonlinear mechanisms.
>
> ### W2: Experiments are mostly on synthetic settings (the only exception is the semi-synthetic GRN example).
>
> Thank you for this comment. We have added experiments on the SERGIO simulator for the causal reasoning task. Please refer to our response to W1&Q2 of Reviewer 1Qqi.
>
> ### Q1: Are synthetic graphs sampled from the prior? How does prior misspecification affect performance?
>
> In the synthetic experiments, the ground-truth graphs and mechanism parameters are sampled directly from the prior. To examine the effect of prior misspecification, we conducted additional experiments on $d = 10$ where the prior assumes an Erdős-Rényi (ER) graph with expected number of edges equal to 2, while the ground-truth graphs are generated from an ER model with expected number of edges equal to 1. Results are shown in Table 1, with a more detailed evaluation provided in Appendix D.6. The GO-CBED-$z$ policy continues to outperform baselines under this mismatch.
>
> __Table 1__: Prior misspecification robustness on $d=10$ ER graphs. Policies trained assuming 2 expected edges per node, evaluated on graphs with 1 edge per node.
>
> | Method      | $\boldsymbol{\mathcal{R}_{T;L}(\pi)} ( \uparrow )$ | $\mathbf{WD} ( \downarrow )$ |
> | ----------- | -------------------------------------------------- | ---------------------------- |
> | GO-CBED-$z$ | $-3.40 \pm 1.79$                               | $0.750 \pm 0.17$         |
> | GO-CBED-$G$ | $-4.40 \pm 0.56$                               | $1.574 \pm 0.28$         |
> | Random      | $-4.82 \pm 0.12$                               | $3.206 \pm 0.81$         |
>
> More broadly, GO-CBED-$z$ remains robust as long as the prior places reasonable mass around the true model (i.e., the ground truth lies within the prior's support). If the prior is made substantially wider, the space of possible trajectories becomes sparser for a fixed training budget, which can lead to a looser variational lower bound and mildly reduced performance. Nevertheless, because our method performs a sequence of experiments, the accumulated data across stages typically provides enough information to contract the posterior toward the ground truth, reducing the impact of moderate prior misspecification. We are happy to run additional experiments for larger graphs ($d=20$) or scale-free families if helpful.

---

> > ### Author Response · Authors · 2025-11-28
> > **Follow-up on Rebuttal**
> >
> > Dear Reviewer `fTBu`,
> >
> > Thank you again for your constructive review.
> >
> > In response to your concerns, we have conducted additional experiments using the SERGIO simulator (`Appendix D.7`) to demonstrate performance under realistic biological settings, as well as robustness evaluations under prior misspecification (`Appendix D.6`).
> >
> > We hope these additions have adequately addressed your concerns. If you find our revisions satisfactory, we would be grateful if you would consider updating your assessment. Otherwise, we remain available to address any remaining questions or concerns.
> >
> > Thank you for your time and valuable insights!
> >
> >
> > Best regards,
> >
> > The Authors of Submission 14524

---

### Author Response · Authors · 2025-11-28
**Summary of Rebuttal and Revisions**

**Dear Reviewers, ACs, and SACs**,

We sincerely thank all reviewers for their thorough and constructive feedback throughout the review process. As the discussion period enters its final week, we would like to briefly summarize the key points from our rebuttal and the improvements we have made in response.

We are encouraged by the reviewers' recognition of our work, including:

* The problem setting is **practically relevant** and **addresses an important gap** in causal experimental design (Reviewers `fTBu`, `cF4q`, `6rGY`).
* The framework is technically **sound** with **clear presentation** and **comprehensive experimental validation** (Reviewers `cF4q`, `6rGY`).
* The goal-oriented, non-myopic formulation represents a **well-motivated advance** over existing greedy or model-centric approaches (Reviewers `cF4q`, `6rGY`).
* Empirical results demonstrate **consistent** performance **gains** across multiple causal tasks (Reviewers `cF4q`, `1Qqi`, `6rGY`).

To address the reviewers' concerns, we have conducted additional experiments and analyses during rebuttal, including:

* SERGIO simulator experiments on causal reasoning tasks (Reviewers `1Qqi`, `fTBu`).
* Prior misspecification robustness tests under graph distribution mismatch (Reviewers `fTBu`, `6rGY`).


**Key revisions in the updated manuscript:**

* Added SERGIO experiments in `Appendix D.7`.
* Added prior misspecification robustness evaluation in `Appendix D.6`.
* Added Wasserstein distance metrics for `Section 5.1` in `Appendix C.2` and corrected `Figure 2` to use the true reward $\mathcal{R}_T(\cdot)$.
* Clarified specification of all causal queries and corresponding ground-truth graphs in `Appendix C.4`.
* Refined statement in `Section 1` to avoid overstating claims.
* Added additional related-work context in `Appendix B`.

All revisions are highlighted in blue. We deeply appreciate the reviewers' constructive feedback and are committed to continually improving our work.

At this stage of the discussion period, we have not yet received comments from the reviewers. We remain available and would be grateful to address any additional questions or concerns to ensure our revisions fully address the reviewers' concerns. Thank you again for your time and valuable insights!

Best regards,

The Authors of Submission 14524

---

### Author Response · Authors · 2025-12-03
**Author Summary**

**Dear Reviewers, ACs, and SACs**,

We sincerely appreciate the constructive feedback provided by the reviewers, and we are especially grateful to the ACs for the additional efforts in evaluating submissions under these challenging circumstances.

**Summary of Initial Reviews**

Reviewers recognized that our problem setting is **practically relevant**, the framework is **technically sound** with **comprehensive experimental validation**, and the goal-oriented formulation represents "a clear shift from model-centric to goal-centric BOED". The main concerns centered on: (1) the need for validation beyond (semi-) synthetic data using biological simulators, (2) robustness to prior misspecification, and (3) experimental clarity.

**Rebuttal Efforts**

During the rebuttal period, we conducted substantial additional experiments to directly address these concerns, including SERGIO simulator experiments for biological validation (`Appendix D.7`), robustness tests under prior misspecification (`Appendix D.6`), Wasserstein distance metrics for downstream evaluation (`Appendix C.2`), and clarified experimental specifications (`Appendix C.4`). We believe these additions comprehensively address the reviewers' concerns. While we did not receive additional feedback from reviewers before the OpenReview incident, we are confident that our revisions directly target the specific issues raised in the initial reviews.

**Core Contributions**

We believe GO-CBED offers meaningful contributions to the causal machine learning community. It is the first framework to unify goal-oriented design with non-myopic sequential planning for causal experimental design, addressing unique challenges including hybrid discrete-continuous uncertainty and interventional action semantics that existing methods do not handle. The framework has direct applications in domains like genomics, where researchers seek to answer specific causal questions rather than recover entire systems. Our comprehensive experiments across synthetic SCMs, DREAM benchmarks, and the SERGIO simulator demonstrate that GO-CBED achieves strong performance on query-focused experimental design tasks, particularly in settings where the downstream objective involves specific causal quantities of interest.

All revisions are highlighted in `blue` in the updated manuscript. For more details, please refer to our `Summary of Rebuttal and Revisions` and responses to individual reviewers.

Thank you again for your time and invaluable service to the community during this period.

With sincere appreciation,

The Authors of Submission 14524

---

### Meta-Review · Area_Chair_Uy5q · 2026-01-07

**Summary:**

Currently borderline paper with no strong support from reviewers.

Reviewer fTBu had concerns about novelty, saying the method puts together already known components. They were also concerned with the fact that experiments are only synthetic. They have also raised an important question on model mismatch between assumed prior and true prior from which model parameters are selected.

Reviewer cF4q had an extensive review, and is similarly concerned about model mismatch. They are also concerned with lack of clarity at several important points (that the correct prior is assumed/used in experiments was not specified). There was a misunderstanding about the use of acyclicity constraint because this was not specified by authors. The reviewer questions sampling acyclic graphs and how slow it is. The reviewer had several other questions most of which were on point, signifying that the paper in its current form suffers from clarity issues.

Reviewer 1Qqi questioned lack of simulator experiments. They also had some issues with over-claiming by authors.

Reviewer 6rGY had several points raised and feedback for the authors some of which overlap with other reviewer concerns such as prior mismatch, runtime/scalability.

**Reviewer Concerns:**

The authors provided an additional simulator experiment to address comments by Reviewer fTBu and Reviewer 1Qqi.

I did not find the authors' experimental result demonstrating model mismatch convincing. It is very limited. The mismatch is the expected number of edges per node in a 10-node graph is 2 in reality whereas it is assumed 1. How about a bigger mismatch? What if the underlying model is not even Erdos-Renyi?

The authors, in their rebuttal to Reviewer cF4q, state that they assume a linear model in their experiments, which is severely limiting. They have indeed used acyclicity constraint but omitted this important detail in the paper. The authors acknowledge that during inference while sampling DAGs they use rejection sampling. This is a fundamental limitation to the scalability of the proposed method.

The authors gave a detailed and extensive reply to Reviewer 6rGY's concerns.

**Reviewer Scores:**

The two supportive reviewers are marginally above acceptance.  I am not sure if they would further increase their scores. It is very likely based on the negative comments by other reviewers, they would be hesitant to do so. I also believe the two borderline acceptance reviews carry sufficient criticism to also justify a reject score. There are clarity issues and very long replies by the authors that the reviewers would most likely request another review cycle.

---

### Decision · Program_Chairs · 2026-01-26

Reject